# Childhood meningitis in rural Gambia: 10 years of population-based surveillance

Usman N. Ikumapayi[1]*, Philip C. Hill[2], Ilias Hossain[1], Yekini Olatunji[1], Malick Ndiaye[1], Henry Badji[1], Ahmed Manjang[1], Rasheed Salaudeen[1], Lamin Ceesay[3], Richard A. Adegbola[4,5], Brian M. Greenwood[6], Grant A. Mackenzie[1,6,7,8]

**1** Medical Research Council Unit, The Gambia at London School of Hygiene & Tropical Medicine, Fajara, The Gambia, **2** Centre for International Health, University of Otago, Dunedin, New Zealand, **3** Ministry of Health, Gambia Government, Banjul, The Gambia, **4** Nigerian Institute of Medical Research, Yaba, Lagos, Nigeria, **5** RAMBICON, Immunisation & Global Health Consulting, Lekki, Lagos, Nigeria, **6** London School of Hygiene & Tropical Medicine, London, United Kingdom, **7** Murdoch Children's Research Institute, Parkville, Melbourne, Australia, **8** Department of Paediatrics, University of Melbourne, Melbourne, Australia

\* Usman-Nurudeen.Ikumapayi@lshtm.ac.uk

## Abstract

### Background

The introduction in many countries of conjugate vaccines against *Haemophilus influenzae* type b, *Streptococcus pneumoniae*, and *Neisseria meningitidis* has led to significant reductions in acute bacterial meningitis (ABM) in children. However, recent population-based data on ABM in sub-Saharan Africa are limited.

### Methods

Population-based surveillance for meningitis was carried out in a rural area of The Gambia under demographic surveillance from 2008 to 2017, using standardised criteria for referral, diagnosis and investigation. We calculated incidence using population denominators.

### Results

We diagnosed 1,666 patients with suspected meningitis and collected cerebrospinal fluid (n = 1,121) and/or blood (n = 1,070) from 1,427 (88%) of cases. We identified 169 cases of ABM, 209 cases of suspected non-bacterial meningitis (SNBM) and 1,049 cases of clinically suspected meningitis (CSM). The estimated average annual incidence of ABM was high at 145 per 100,000 population in the <2-month age group, 56 per 100,000 in the 2–23-month age group, but lower at 5 per 100,000 in the 5–14-year age group. The most common causes of ABM were *Streptococcus pneumoniae* (n = 44), *Neisseria meningitidis* (n = 42), and Gram-negative coliform bacteria (n = 26). Eighteen of 22 cases caused by pneumococcal serotypes included in PCV13 occurred prior to vaccine introduction and four afterwards. The overall case fatality ratio for ABM was 29% (49/169) and was highest in the <2-month age group 37% (10/27). The case fatality ratio was 8.6% (18/209) for suspected non-bacterial meningitis and 12.8% (134/1049) for clinically suspected meningitis cases.

**Data Availability Statement:** The Data cannot be shared publicly as they hold potential sensitive information regarding the participants. Data are available from Gambian Government / Medical Research Council (UK) Joint Ethics Committee.

Data can be requested to the ethics committee at "Ethics@mrc.gm" by researchers who meet the criteria for access to confidential data. All other relevant data are presented within the article.

**Funding:** This work was supported by the Bill & Melinda Gates Foundation (grant number OPP1020327) and the Medical Research Council Unit The Gambia at London School of Hygiene and Tropical Medicine." The grant recipient was Dr Grant A Mackenzie.

**Competing interests:** The authors have declared that no competing interests exist.

## Conclusions

Gambian children continue to experience substantial morbidity and mortality associated with suspected meningitis, especially acute bacterial meningitis. Such severely ill children in sub-Saharan Africa require improved diagnostics and clinical care.

## Introduction

Acute bacterial meningitis (ABM) is an infectious disease syndrome characterised by bacterial infection and inflammation of the meninges and is associated with high mortality and morbidity [1,2]. ABM is among the major killers of children living in the meningitis belt of sub-Sahara Africa [3], which has historically experienced high rates of endemic and epidemic meningitis due to *Neisseria meningitides* [4,5] *Haemophilus influenzae* type b (Hib) [6] and *Streptococcus pneumoniae* [7]. The recent introduction of conjugate vaccines in many African countries has led to some reductions in the incidence of meningitis due to Hib, *S. pneumoniae* and group A *N. meningitides* [7–9]. Given that most meningitis studies are hospital-based, there are limited population-based data on the incidence of meningitis in sub-Saharan Africa [10]. Furthermore, data on the incidence and outcome of suspected non-bacterial meningitis in Africa are scarce.

In The Gambia, Hib and pneumococcal conjugate vaccines (PCV) were introduced in May 1997 (Hib), August 2009 (PCV7) and May 2011 (PCV13) respectively with high coverage (>90%). In 2013, individuals aged 1–29 years were vaccinated nationwide against the group A meningococcus [11]. Population-based surveillance has shown a substantial impact of the introduction of PCVs with an 80% reduction in the incidence of invasive pneumococcal diseases (IPD), in the under-5-year age group [12]. In this study, we used 10 years of standardised population-based surveillance for meningitis to describe incidence of meningitis over time, its aetiology, and case-fatality among children with ABM in a rural part of The Gambia in the era of widespread deployment of meningitis related vaccines (Hib, PCV and Men-A).

## Materials and methods

### Study site and participants

The Pneumococcal Surveillance Program (PSP) has conducted population-based surveillance for invasive bacterial disease in the Upper River Region of The Gambia since 2008 [13]. The PSP was based at the Basse Field Station of the Medical Research Council Unit The Gambia at the London School of Hygiene & Tropical Medicine (LSHTM).

We conducted surveillance for suspected pneumonia, sepsis and meningitis between May 12, 2008, and December 31, 2017. The population included all residents of the Basse Health and Demographic Surveillance System (BHDSS). The population was enumerated every 4 months, with births, deaths, migrations, and vaccinations been recorded. The estimated population in 2017 was 183,946 of whom 35,597 (20%) were younger than 5 years of age.

### Surveillance procedures

Nurses used standardized criteria to screen all patients aged 2–59 months who presented at a health facility in the surveillance area for suspected septicaemia, pneumonia or meningitis, while infants in the first 2 months of life were screened only for suspected meningitis. Children who were positive on screening were referred to a clinician who used standardized criteria to

determine a surveillance diagnosis and who initiated a standardized programme of investigation [14]. Meningitis in patients aged from 0–59 months of age was defined according to clinical judgement and suspected if one of the following was present: neck-stiffness, impaired consciousness, prostration, history of convulsion, or a bulging fontanelle. Among those aged 5 years and above, suspected meningitis was defined according to clinical judgement and considered if any of the following were present: axillary temperature $\geq$38˚C and meningism (neck stiffness) and/or photophobia or altered mental state (Glasgow Coma Score <14). Patients with suspected meningitis had blood and cerebrospinal fluid (CSF) samples taken for culture. Inclusion criteria for this analysis were age $\leq$14 years and meeting the definition for suspected meningitis. Exclusion criteria were non-residence in the study area, trauma or a suspected hospital-acquired infection.

The definition of acute bacterial meningitis (ABM) was a CSF leucocyte count >5cells/mm$^3$, and a positive culture of CSF or blood in a patient with clinical signs of meningitis. The definition for clinically suspected meningitis (CSM) cases was the presence of clinical features of meningitis, as described above, but a normal CSF leucocyte count 0-5cells/mm$^3$ and a negative culture and antigen test. Suspected non-bacterial meningitis (SNBM), commonly termed aseptic meningitis, was defined as a CSF leukocyte count >5cells/mm$^3$, a CSF negative antigen and a negative culture of CSF and blood in a patient with clinical signs of meningitis.

## Outcomes

The primary outcome of the study was the incidence of acute bacterial and suspected non-bacterial meningitis during the study period. Secondary outcomes were the aetiology of ABM and the case fatality ratio in CSM, ABM, and SNBM groups.

## Laboratory methods

Blood only or CSF only, or both blood and CSF, were collected from patient for conventional microbiological investigations. A 3 to 5 ml venous blood sample was collected from study participants. Lumbar puncture was performed to obtain 1 to 2 ml CSF. Blood was inoculated into BACTEC bottles (Becton Dickinson) and incubated in an automated BACTEC 9050 blood culture system (Becton Dickinson, Erembodegem, Belgium) for a maximum of 5 days. Positive cultures were sub-cultured on blood and chocolate agar and examined for bacterial growth following 24 or 48-hours of aerobic incubation, incubation in 5–10% $CO_2$ and anaerobic incubation, all at 37˚C. Similarly, CSF was cultured on blood and chocolate agar and isolates identified by biotyping, and serotyping for pneumococcal serotypes with a latex agglutination assay using factor and group-specific antisera (Statens Serum Institut, Copenhangen, Denmark). *Neisseria meningitidis* isolates' serogroup were identified by latex agglutination using Ramel (Thermo Fisher Scientific, Waltman, MA, USA) test kits. *Haemophilus influenzae* type b and other encapsulated *H. influenzae* other than type b isolates were distinguished by slide latex agglutination using polyvalent and monovalent *Haemophilus influenzae* Difco serotyping antisera to Hi of types a, b, c, d, e and f (Beckton Dickinson, Difco, Belgium). In line with the surveillance procedures and WHO guidelines (WHO/CDS/CSR/EDC/99.7) presumptive diagnoses that include; cell count, Gram-stain and bacterial antigen (latex agglutination) tests were routinely performed on CSF whilst waiting for a culture confirmatory result. Rapid diagnostic tests for malaria using ICT-Malaria *p.f.* Antigen tests (ICT Diagnostics, Cape Town, South Africa) were performed routinely during the malaria transmission season. Sixty-seven children with clinical signs of meningitis but who had a positive rapid ICT diagnostic test for malaria and normal leukocyte count were excluded from our analysis.

### Antibiotic susceptibility testing

Disk diffusion methods were used to detect antibiotic resistance following standard guidelines [15].

### Statistical analysis

Analyses were restricted to residents of the BHDSS. We calculated overall and age-stratified incidence for each category of patients during the observation period using overall and age-stratified numbers of cases divided by the overall and age-stratified BHDSS population denominators. We investigated trends in incidence over time, calculating annual incidence with mid-point population denominators from the BHDSS. We calculated disease incidence for three different categories of patient: CSM, ABM and SNBM. Calculations of incidence and 95% confidence intervals assumed a Poisson distribution were done using Stata 14 software package. We calculated case-fatality ratios (CFR) for each of the three categories of patient CSM, ABM and SNBM based on the number of cases who died during and after hospital admission divided by the total number of cases in each category.

The period prior to the introduction of PCV13 (pre-PCV13) is defined as May 12, 2008 to December 31, 2012, whilst the period after the introduction of PCV13 (post-PCV13) is defined as January 1, 2013 until December 31, 2017. A p-value $<0.05$ was considered statistically significant.

### Ethical considerations

The study was approved by the Gambia Government / Medical Research Council (UK) Joint Ethics Committee (number SCC 1087). Parent or guardian of participants gave written, informed consent for participation in the study (Pneumococcal Surveillance Program).

## Results

### Demographic and clinical characteristics

Over the almost 10-year period of surveillance, 26,408 children were enrolled in PSP and 1,666 (6.3%) were referred by nurses for clinical review as possible cases of meningitis. Of those with clinical signs of meningitis, 14.3% (239/1666) were excluded from the analysis. Of these, 72.0% (172/239) were due to blood and CSF not being taken for culture as the children were severely ill and dehydrated. 7.1% (17) of these cases died and, 9.6% (23) were referred prior to sample collection. The other 28% (67/239) were cases of clinical sign meningitis that were positive rapid ICT test for malaria, and there were 5 (7.4%) deaths from this group. Blood or CSF, or both, taken from 85.7% (1427/1666) of the suspected cases with clinical signs of meningitis were considered in the analysis (Fig 1). The total number of blood cultures and CSF cultures performed were 1,070 and 1,121 respectively (Fig 1). Blood and CSF were obtained from 764 children, blood only from 306, and CSF only from 357. All bacterial antigen positive CSFs samples were also culture positive although some CSF had very scanty growth ($<20$ x $10^5$ organisms/mL). All bacterial antigen negative CSF were culture negative, although only 87% of cultured CSF samples were tested for bacterial antigens. Approximately, 12% (169/1427) of suspected meningitis cases were bacterial antigen and culture positive, whilst bacterial antigen and culture negative cases accounted for 85% (1217/1427) and contaminants for a further 3% (41/1427) (Fig 1). The organisms considered contaminants in the CSF were *Streptococcus viridians*

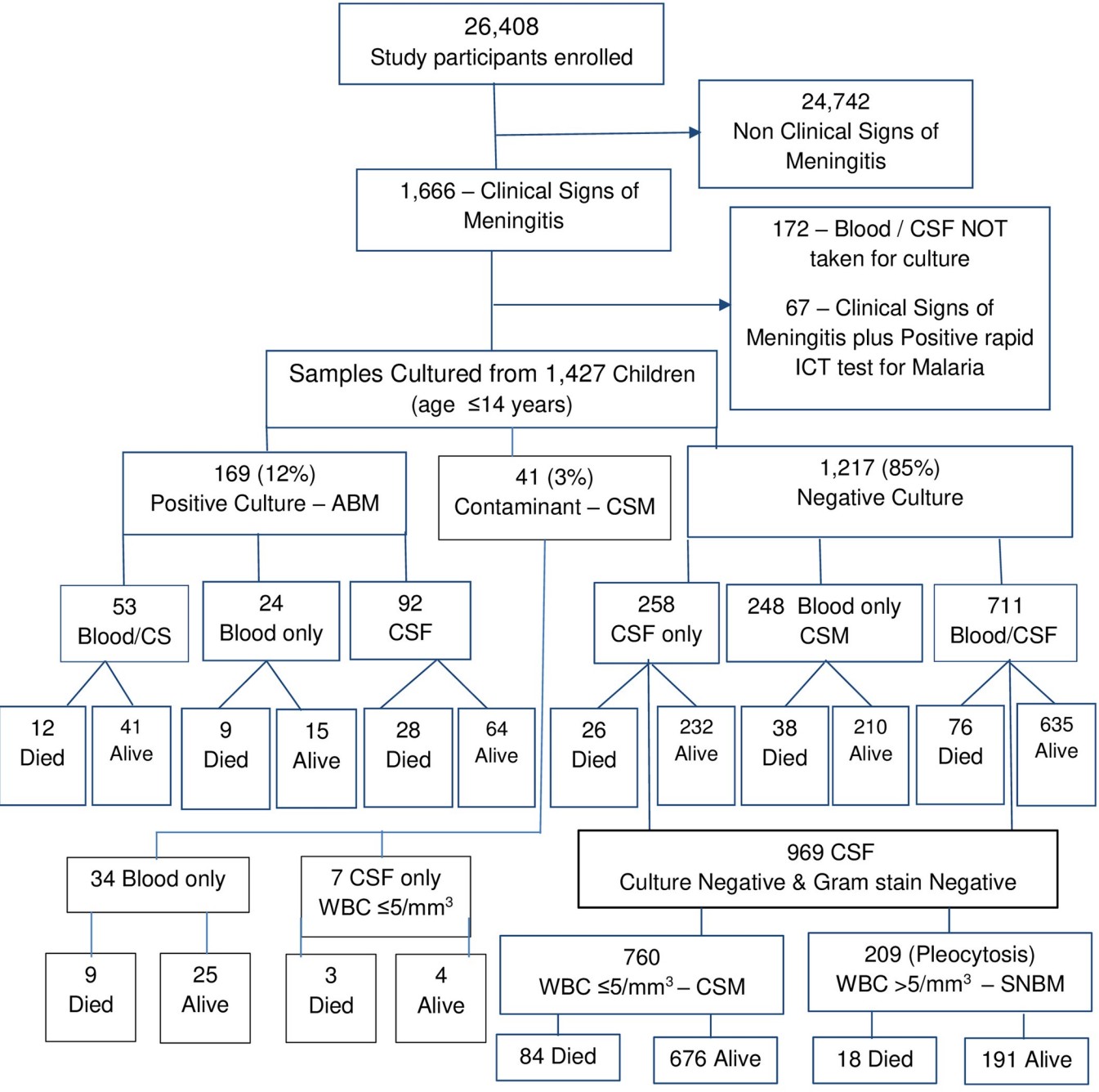

**Fig 1. Enrolment, diagnoses and outcome.**

**Table 1. Baseline characteristics of meningitis in rural Gambia: 10 years of population-based surveillance.**

| Characteristic | Category | Acute Bacterial Meningitis (n = 169) N (%) | Clinically Suspected Meningitis (n = 1049) N (%) | Suspected Non-Bacterial Meningitis (n = 209) N (%) |
|---|---|---|---|---|
| Gender | Male | 90 (53.3) | 624(59.5) | 110 (52.6) |
| | Female | 79 (46.7) | 425 (40.5) | 99 (47.4) |
| Age in months | <2 | 27 (16.0) | 115 (11.0) | 29 (13.9) |
| | 2–23 | 65 (38.4) | 405 (38.6) | 83 (39.7) |
| | 24–59 | 50 (29.6) | 439 (41.8) | 70 (33.5) |
| | 60–168 | 27 (16.0) | 90 (8.6) | 27 (12.9) |

5/7 and coagulase negative Staphylococcus (CNS) 2/7. In the blood, contaminants were *Streptococcus viridans* 8/34, Micrococcus 6/34, CNS 12/34 and Bacillus species 8/34.

The ABM, CSM and SNBM cases were more likely to be male than female. (Table 1). The ABM and SNBM were proportionately higher in those aged 2-23-months compared to other age groups, whereas those aged 24-59-months predominated for CSM. However, there were no statistically significant differences by age group (Table 1). The additional age stratum that was not separately presented in figure and in table was neonatal age 0-4-week that account for 141/1427. Of these, there were 41 deaths of which ABM, SNBM and CSM account for 7/41, 4/41 and 30/41 respectfully.

There were 14 children under 2 months of age who were in the ABM category and had CSF culture and cell count performed, and all had CSF leukocyte counts >10 cells/mm$^3$.(range 45 to 1240cells/mm$^3$).

## Incidence

The incidence of ABM during the study period in all age groups combined was 20.3 per 100,000 (95% CI 17–24) (Table 2A and 2B) with a peak of 63.3 per 100,000 (95% CI 48–83) in 2012 (Table 2A). The incidence of ABM and SNBM varied from year to year, while that of CSM was relatively stable over time.

## Case fatality

Mortality due to ABM showed statistical significance [OR 2.97, 95% CI 1.99–4.37), p-value <0.001) compared to CSM and SNBM (Table 3). The annual case fatality ratio for ABM varied considerably from year to year whilst the case fatality ratio for CSM and SNBM was relatively constant over time (Fig 2). The overall case fatality was 29% (49/169) for ABM, 8.6% (18/209) for SNBM and 12.8% (134/1049) for CSM. Mortality was highest in the first 2 months of life (37%), but also high in other age groups; 29%, 30%, and 18% in the 2–23 months, 2–4 years, and 5–14 years age strata respectively (Fig 3). In all age strata, case fatality was lower for clinically suspected but not proven meningitis and for SNBM compared to ABM (Fig 3). Our study also showed that 100% (49/49) of the ABM related death occurred within the first 5 days of hospitalization. Whilst SNBM and CSM related death occurred within 3 and 15 days of hospitalization respectively. The number of deaths following hospital discharge for SNBM and CSM were 2/18 and 9/134 respectively.

## Seasonality

The three categories of clinical signs of meningitis were proportionately higher during the dry season (November-June) compared to the wet season (July-October) (Fig 4).

**Table 2.** a: Incidence per 100,000 population of clinically suspected meningitis, suspected non-bacterial meningitis and acute bacterial meningitis among children ≤14 years of age (2008–2017), by year (n = 1427). b: Incidence per 100,000 population of clinically suspected meningitis, suspected non-bacterial meningitis and acute bacterial meningitis among children ≤14 years of age (2008–2017), by age (n = 1427).

| Year | Clinically Suspected Meningitis (n = 1,049) | | | Suspected Non-Bacterial Meningitis (n = 209) | | | Acute Bacterial Meningitis (n = 169) | | |
|---|---|---|---|---|---|---|---|---|---|
| | Cases | Incidence | 95% CI | Cases | Incidence | 95% CI | Cases | Incidence | 95% CI |
| **2008** | 97 | 279.5 | 227–341 | 12 | 34.6 | 18–60 | 8 | 23.1 | 10–45 |
| **2009** | 126 | 167.0 | 139–199 | 1 | 1.3 | 03–07 | 12 | 15.9 | 8–28 |
| **2010** | 122 | 154.7 | 128–185 | 13 | 16.5 | 09–28 | 16 | 20.3 | 12–33 |
| **2011** | 155 | 189.6 | 161–222 | 29 | 35.5 | 24–51 | 18 | 22.0 | 13–35 |
| **2012** | 101 | 118.4 | 96–143 | 35 | 41.0 | 29–57 | 54 | 63.3* | 48–83 |
| **2013** | 89 | 102.4 | 82–126 | 30 | 34.5 | 23–49 | 13 | 15.0 | 8–26 |
| **2014** | 112 | 127.1 | 105–153 | 22 | 24.9 | 16–38 | 23 | 26.1 | 17–39 |
| **2015** | 103 | 117.5 | 96–143 | 39 | 44.5 | 32–61 | 17 | 19.4 | 11–31 |
| **2016** | 77 | 87.4 | 69–109 | 17 | 19.3 | 11–31 | 4 | 4.5 | 1.2–12 |
| **2017** | 67 | 75.5 | 58–96 | 11 | 12.4 | 06–22 | 4 | 4.5 | 1.2–12 |
| **Total** | 1049 | 125.9 | 118–134 | 209 | 25.1 | 22–29 | 169 | 20.3 | 17–24 |
| **Age in Month** | Clinically Suspected Meningitis (n = 1,049) | | | Suspected Non-Bacterial Meningitis (n = 209) | | | Acute Bacterial Meningitis (n = 169) | | |
| | Cases | Incidence | 95% CI | Cases | Incidence | 95% CI | Cases | Incidence | 95% CI |
| **<2** | 115 | 616.6 | 509–740 | 29 | 155.5 | 104–223 | 27 | 144.8 | 95–210 |
| **2–23** | 405 | 345.7 | 313–381 | 83 | 70.8 | 56–87 | 65 | 55.5 | 43–71 |
| **24–59** | 439 | 227.5 | 207–249 | 70 | 36.3 | 28–46 | 50 | 25.9 | 19–34 |
| **60–168** | 90 | 17.8 | 14–22 | 27 | 5.4 | 04–08 | 27 | 5.4 | 4–8 |
| **Total age** | 1049 | 125.9 | 118–134 | 209 | 25.1 | 22–29 | 169 | 20.3 | 17–24 |

Note: Only 234 days of surveillance in 2008, from 12 May– 31 Dec.

*Higher incidence due to the epidemic of *Neisseria meningitidis* W135.

## Aetiology

Pathogenic bacteria were isolated from 77 blood cultures and from 92 CSF specimens from patients with a raised WBC count (Table 4). *S. pneumoniae* was isolated in 44 cases. *H. influenzae* type-b was found in 12 cases, 4 of whom had been vaccinated against Hib. There were 42 cases of meningococcal meningitis, all belonging to serogroup W, with 35 in 2012 during a W135 epidemic. Other common causes were Gram negative coliforms (n = 26), *Staphylococcus aureus* (n = 16) and non-typhoidal Salmonella (n = 12), non-type b *Haemophilus influenzae* (n = 6; 5/6 of these were *Haemophilus influenzae* type a), *Klebsiella pneumoniae* (n = 6) and *Escherichia coli* (n = 5).

*S. pneumoniae* was categorized into vaccine (n = 22) and non-vaccine (n = 22) serotypes based on the serotypes included in PCV13 (Table 5). We detected six vaccine serotypes (1, 5,

**Table 3. Fatal outcome of meningitis in rural Gambia (2008–2017).**

| | Fatal outcome | | | | |
|---|---|---|---|---|---|
| | **Died (n = 201)** N (%) | **Survived (n = 1,226)** N (%) | **Total (n = 1,427)** N (%) | **OR (95% CI)** | **P value** |
| **ABM** | 49 (24.4) | 120 (9.8) | 169 (11.84) | 2.97 (1.99–4.37) | <0.001 |
| **CSM** | 134 (66.6) | 915 (74.6) | 1,049 (73.51) | 0.68 (0.49–0.95) | 0.017 |
| **SNBM** | 18 (9.0) | 191 (15.6) | 209 (14.65) | 0.53 (0.30–0.89) | 0.013 |

**Key:** ABM–Acute Bacterial Meningitis, CSM–Clinically Suspected Meningitis and SNBM–Suspected Non-Bacterial Meningitis.

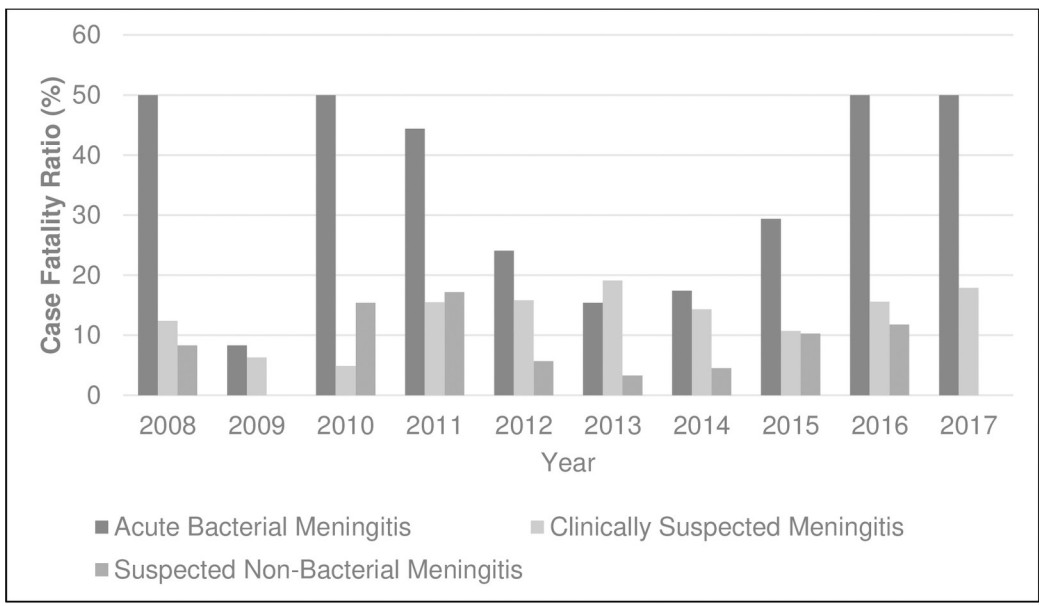

**Fig 2. Annual Case Fatality Ratio of Clinically Suspected Meningitis (CSM), Suspected Non-Bacterial Meningitis (SNBM) and Acute Bacterial Meningitis (ABM) among children aged 1 day -14 years in Upper River Region Gambia, 2008–2017.**

14, 6A, 19F and 23F). Eighty-two percent (18/22) of cases with vaccine serotypes were detected in the pre-PCV13 period compared to 18% (4/22) after the introduction of PCV13, one of whom had been vaccinated. We detected 15 non-vaccine serotypes (2, 21, 46, 9A, 10F, 12B, 12F, 15A, 15B, 16F, 17F, 18A, 23B, 25F and 35B), a half (11/22) of the cases infected with one

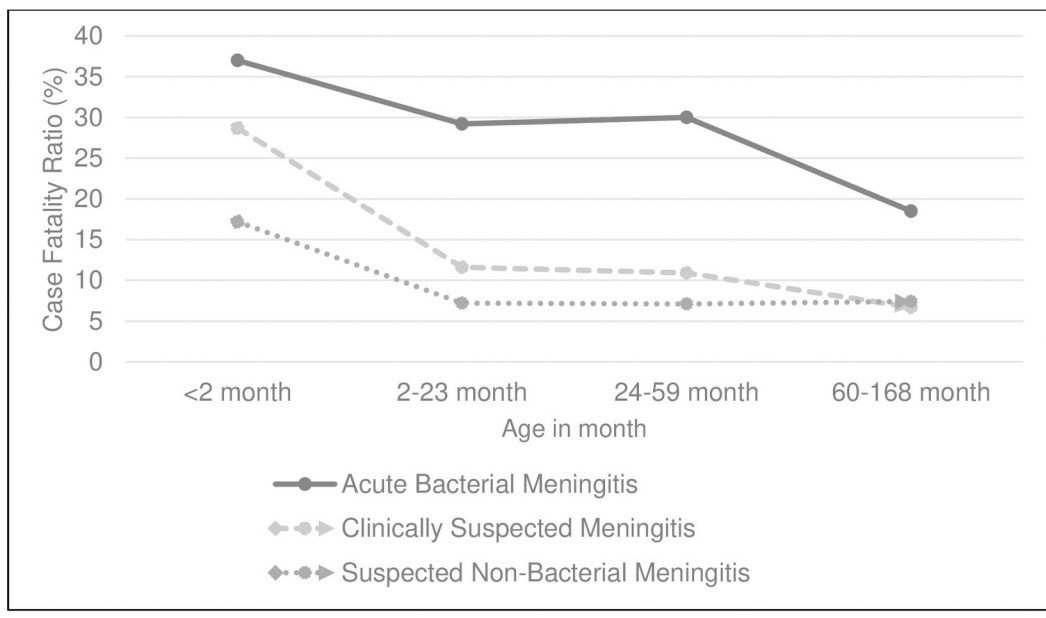

**Fig 3. Age Strata Case Fatality Ratio of Clinically Suspected Meningitis (CSM), Suspected Non-Bacterial Meningitis (SNBM) and Acute Bacterial Meningitis (ABM) among children aged 1 day -14 years in Upper River Region Gambia, 2008–2017.**

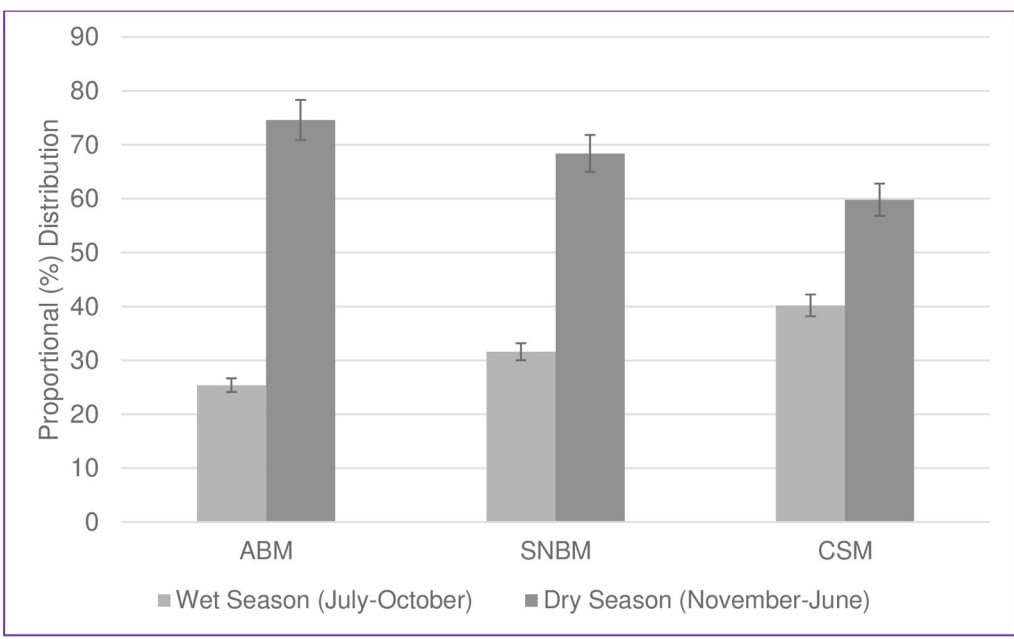

**Fig 4. Seasonal Distribution of Clinically Suspected Meningitis (CSM), Suspected Non-Bacterial Meningitis (SNBM) and Acute Bacterial Meningitis (ABM) Over 10 Years in Upper River Region Gambia, 2008–2017.**

of these were detected pre and post PCV 13 introduction respectively. In the pre-PCV13 era, 6/7 pneumococcal meningitis deaths were attributed to vaccine serotypes compared to 1/7 pneumococcal meningitis death in the post-PCV13 era. In the pre-PCV13 era, there were 5/8 non-vaccine serotype pneumococcal deaths compared to 3/8 non-vaccine serotype pneumo-coccal meningitis deaths in post-PCV13 era (Table 5). Also, in the pre-PCV13 era, among the children aged 5-24-month, there were 4/6 vaccine serotype pneumococcal meningitis deaths compared to 4/5 non-vaccine serotype pneumococcal meningitis deaths. In post-PCV13 era, the only pneumococcal meningitis death caused by vaccine serotype 23F and the child was aged 24-month. In the post-PCV13 era, there were three non-vaccine serotype (serotype 2,

**Table 4. Frequency of bacteria isolated from blood and CSF cultured and corresponding case fatality ratio (CFR) caused among children in Upper River Region Gambia, 2008–2017 (n = 169).**

| Bacteria | Total N | Blood N | CSF N | No. of Death | CFR % | 95% CI |
|---|---|---|---|---|---|---|
| **All Bacteria** | 169 | 77 | 92 | 49 | 29 | 21.4% - 38.3% |
| *Streptococcus pneumoniae* | 44 | 16 | 28 | 15 | 34.1 | 19.1% - 56.2% |
| *Neisseria meningitidis* W | 42 | 10 | 32 | 9 | 21.4 | 9.8% - 40.7% |
| Other GNR* | 26 | 20 | 6 | 5 | 19.2 | 6.2% - 44.9% |
| *Staphylococcus aureus* | 16 | 13 | 3 | 6 | 37.5 | 13.8% - 81.6% |
| *H. influenzae* type b | 12 | 1 | 11 | 3 | 25 | 5.2% - 73.1% |
| Non-Typhoidal Salmonella | 12 | 8 | 4 | 4 | 33.3 | 9.1% - 85.3% |
| non-type b *H. influenzae* | 6 | 1 | 5 | 1 | 16.7 | 0.4% - 92.9% |
| *Klebsiella pneumoniae* | 6 | 6 | 0 | 4 | 66.7 | 18.2% -170.7% |
| *Escherichia coli* | 5 | 2 | 3 | 2 | 40 | 4.8% - 144.5% |

*GNR–Gram Negative Rods that include *Pseudomas luteola*, *Pseudomanas stuteria*, *Pseudomonas floreense*, *Serratia marcenscens*, *Chromosoma violacum*, *Enterococcus faecalis*, *Stentrophomonas maltophilia*

**Table 5. Distribution of twenty-one pneumococcal serotypes causing pneumococcal meningitis (n = 44).**

| Vaccine Serotype (n = 22) | | | |
|---|---|---|---|
| Serotype (n = 6) | Pre-PCV13 N (%) [No. Dead] | Post-PCV13 N (%) [No. Dead] | Total N (%) [No. Dead] |
| 1 | 4 (18.2) [1] | 2 (9.0) [0] | 6 (27.3) [1] |
| 5 | 4 (18.2) [1] | 0 (0) | 4 (18.2) [1] |
| 14 | 4 (18.2) [1] | 0 (0) | 4 (18.2) [1] |
| 6A | 2 (9.0) [1] | 0 (0) | 2 (9.0) [1] |
| 19F | 2 (9.0) [1] | 0 (0) | 2 (9.0) [1] |
| 23F | 2 (9.0) [1] | 2 (9.0) [1] | 4 (18.2) [2] |
| **Total** | **18 (82) [6]** | **4 (18) [1]** | **22 (100) [7]** |

| Non-Vaccine Serotype (n = 22) | | | |
|---|---|---|---|
| Serotype (n = 15) | Pre-PCV13 N (%) [No. Dead] | Post-PCV13 N (%) [No. Dead] | Total N (%) [No. Dead] |
| 2 | 0 (0) | 2 (9.0) [1] | 2 (9.0) [1] |
| 21 | 0 (0) | 1 (4.5) | 1 (4.5) |
| 46 | 1 (4.5) | 0 (0) | 1 (4.5) |
| 9A | 1 (4.5) [1] | 0 (0) | 1 (4.5) [1] |
| 10F | 0 (0) | 1 (4.5) | 1 (4.5) |
| 12B | 2 (9.0) [1] | 0 (0) | 2 (9.0) [1] |
| 12F | 2 (9.0) [2] | 1 (4.5) | 3 (13.6) [2] |
| 15A | 1 (4.5) | 0 (0) | 1 (4.5) |
| 15B | 1 (4.5) | 0 (0) | 1 (4.5) |
| 16F | 1 (4.5) | 0 (0) | 1 (4.5) |
| 17F | 1 (4.5) [1] | 1 (4.5) | 2 (9.0) [1] |
| 18A | 0 (0) | 1 (4.5) [1] | 1 (4.5) [1] |
| 23B | 0 (0) | 1 (4.5) | 1 (4.5) |
| 25F | 0 (0) | 2 (9.0) [1] | 2 (9.0) [1] |
| 35B | 1 (4.5) | 1 (4.5) | 2 (9.0) |
| **Total** | **11 (50) [5]** | **11 (50) [3]** | **22 (100) [8]** |

**NB:** Pre-PCV13 vaccine is defined as occurrence of cultured confirmed pneumococcal meningitis from May 12, 2008, to December 31, 2012. Whilst post-PCV13 is defined as occurrence of cultured confirmed pneumococcal meningitis from January 1, 2013, until December 31, 2017.

18A and 25F) pneumococcal meningitis deaths in children aged 7 days, 28 days, and 36-month (Table 5).

## Bacterial resistance patterns

All pneumococcal isolates were susceptible to penicillin, ampicillin, and cefotaxime with 64% resistant to co-trimoxazole and 14% resistant to chloramphenicol (Table 6). All meningococcal isolates were susceptible to cefotaxime. Resistance among non-type b *H. influenzae* isolates was 8% for cefotaxime, 17% for ampicillin, and 33% for chloramphenicol. *Escherichia coli*, *Klebsiella pneumoniae*, and non-typhoidal Salmonella (NTS) were generally sensitive to cefotaxime, ciprofloxacin and gentamicin but not ampicillin. There were no significant different in the rate of pneumococcal resistance against many antibiotics before introduction of PCV13 compared with post-PCV13. However, the rate of pneumococcal resistance against chloramphenicol and erythromycin was over 100% higher post-PCV13 compared to pre-PCV13.

**Table 6. Bacterial antimicrobial resistance patterns against nine antibiotics.**

| Bacteria | Number of Isolates | Antimicrobial Resistance, n (%) | | | | | | | | |
|---|---|---|---|---|---|---|---|---|---|---|
| | | AMP N (%) | CTX N (%) | CHL N (%) | CIP N (%) | SXT N (%) | ERY N (%) | PEN N (%) | TET N (%) | CN N (%) |
| *S. pneumoniae* | 44 | 0 (0) | 0 (0) | 6 (14) | 12 (27) | 28 (64) | 4 (9) | 0 (0) | 18 (41) | N/A |
| *N. meningitidis* | 42 | 5 (12) | 0 (0) | 3 (7) | 0 (0) | 27 (64) | 2 (5) | 2 (5) | 4 (10) | N/A |
| Other GNR | 26 | 7 (27) | 4 (15) | 5 (19) | 3 (12) | 9 (35) | N/A | N/A | 6 (23) | 8 (31) |
| *Staphylococcus aureus* | 16 | *OX 5 (31) | N/A | 0 (0) | N/A | 6 (38) | 2 (13) | 12 (75) | 4 (25) | 3 (18) |
| *H. influenzae* type b | 12 | 2 (17) | 1 (8) | 4 (33) | 0 (0) | 8 (67) | 6 (50) | 3 (25) | 4 (33) | N/A |
| NTS | 12 | 5 (42) | 0 (0) | 1 (8) | 1 (8) | 1 (8) | N/A | N/A | 1 (8) | 1 (8) |
| *non-type b H. influenzae* | 6 | 0 (0) | 0 (0) | 3 (50) | 0 (0) | 0 (0) | 1 (17) | 0 (0) | 3 (50) | N/A |
| *K. pneumoniae* | 6 | 1 (17) | 2 (33) | 1 (17) | 0 (0) | 1 (17) | N/A | N/A | 0 (0) | 0 (0) |
| *Escherichia coli* | 5 | 2 (40) | 0 (0) | 2 (40) | 0 (0) | 1 (20) | N/A | N/A | 1 (20) | 2 (40) |

**Key:** AMP Ampicillin, CTX Cefotaxime, CHL Chloramphenicol, CIP Ciprofloxacin, SXT Cotrimoxazole, ERY Erythromycin, PEN Penicillin, TET Tetracycline, CN Gentamycin and OX Oxacillin, GNR Gram-negative rod, N/A Not Applicable and GNR–Gram Negative Rod.

**Note:** Disk diffusion methods were used following standard guidelines (CLSI 2012, M100-S22, Vol. 32 No.3).

## Discussion

This paper provides new information on meningitis incidence and aetiology in The Gambia during a period after the introduction of Hib and pneumococcal conjugate vaccine 3 plus 0 schedules. Over the study period, there was no considerable decrease in the incidence of ABM except in 2016 and 2017 which may be due to low rainfalls resulted in low malaria transmission with a subsequent declined in cerebral malaria presentation.

The sharp increased seen in 2012 was due to an outbreak caused by *N. meningitidis* serogroup W [16], as was the case in other parts of the meningitis belt in 2012 and 2015 [17]. The gradual reduction in the incidence of ABM was likely due to the introduction of vaccination strategies targeting the three most common bacterial causes of meningitis—*H. influenzae* type-b [18], *N. meningitidis* [19,20] and *S. pneumoniae* [12,21]. However, the annual incidence of clinically suspected but not confirmed and SNBM cases remained relatively stable from 2008 to 2017 with a mean incidence of 141.9, per 100,000 and 26.45, per 100,000 respectively. Few studies in sub-Saharan Africa have reported on cases of SNBM [22]. However, an earlier study reported the incidence of clinically suspected meningitis was greatest in 2011, potentially related to an increase in presentation of convulsions that may be associated with cerebral malaria and a high level of malaria transmission following widespread flooding in 2010 [14].

Despite progress in decreasing the mortality caused by ABM in the Gambia, there were still many childhood deaths due to ABM in this rural study, with an overall CFR for ABM of 29% and a CFR of 12.8% for CSM and 8.6% for SNBM cases. A similar study in Taiwan showed a CFR 10.9% (6/56) among ABM cases but a CFR of 0% (0/141) among SNBM cases [23]. It is unclear whether deaths observed among SNBM children in our region were due to infection with other microbial agents, or whether these children had encephalitis as well as meningitis with the former having high mortality. It may also have been due to few cases of missed diagnosis cause by prior-antibiotic use before lumber-puncture. Although the common practice in our formal health system is to obtain LP prior antibiotic usage but antibiotics are available in the study area outside the formal health system. Thus, few cases might have taken antibiotic before presenting to the hospital.

The annual CFR for ABM compared to CSM and SNBM was consistently higher throughout the study period. The reason for the particularly high fatality in years 2008, 2010, 2011, 2016 and 2017 is unclear. This observation was also made in earlier studies from Africa and Europe describing the high mortality of bacterial vaccine-preventable diseases [7,24]. Many previous studies have reported a high CFR for ABM [25] but few have shown a significant level of CFR in CSM and SNBM cases.

Both the incidence of ABM and its case fatality ratio were highest in the youngest age group in keeping with other studies [16,26], perhaps due in part to the fact that these children had not been vaccinated; a high incidence and CFR from SNBM group was seen also among the youngest children. There are limited recent studies from West-Africa that report incidence and CFR of CSM and SNBM with which to compare our findings [27,28]. There was no statistically significant difference of ABM between genders although, the incidence of ABM was proportionately higher among males compared to females, as reported in a similar study in Brazil [26]. A concern is the high incidence in cases of SNBM which can cause clinical diagnostic uncertainty, empirical-antibiotic-treatment and undirected patient management. In-turn, this can result in a poor patient outcome, increased risk of child disability [29–31] and a further burden to hospital resources and society [32].

The ABM, SNBM and CSM were consistently high in the dry season compared to the wet season. ABM was higher compared to SNBM and CSM in dry season, but in the wet season CSM was higher compared to ABM and CSBM. This observation is consistent with other studies from sub-Saharan Africa associating meningitis with dry weather [16,27,33].

The number of cases of *S. pneumoniae* meningitis due to vaccine serotype decreased substantially 12 months after the introduction of PCV13 vaccination from 82% to 18%. Concordantly, higher proportion of pneumococcal meningitis deaths (85.7%) attributed to vaccine serotype in the pre-PCV13 era highlight the impact of the introduction of PCV13 reported in the past studies [9,12]. Nevertheless, *S. pneumoniae* was the most common cause of ABM with vaccine serotype and non-vaccine serotype accounting for 50% (22/44) each.

The only pneumococcal meningitis death by vaccine serotype in post-PCV13 vaccination was aged 24 months caused by serotype 23F in August 2013, although the child may have died from immunocompromised complications because the mother was HIV tested positive after the death of the child. The two pneumococcal meningitis deaths in non-vaccine serotypes (serotype 2 and 18A) in post-PCV13 were of ages 7 days and 28 days in July and November 2017 respectively, and one pneumococcal meningitis death age 36-months caused by serotype 25F in February 2014 emphasized the need to introduce high valency PCV into the country's Extended Immunization Program (EPI). Cases of meningitis due to non-vaccine serotypes remained relatively constant in pre-PCV13 and post-PCV13 which strengthens the evidence that the reduction in cases of vaccine serotype meningitis seen after the introduction of PCV was due to the vaccine, as found in earlier studies [12]. Immunisation with Hib conjugate vaccine was introduced into routine EPI programme of The Gambia in 1997 and from 2008 to 2010 there were no bacterial meningitis cases due to Hib in the study area. However, there were sporadic Hib cases from 2011 to 2013 and 50% of all the Hib cases were identified in 2013 and in 2013, a study undertaken in the same region from 2010 to 2013 reported an incidence of Hib disease of 88 (95% CI 29–207) per 100,000 in those aged 2–11 month and 22 (95% CI 9–45) per 100,000 in those aged 2–59 months. The reason for this sudden rise in the incidence of Hib was unexplained but cases included a few vaccine failures and many cases in babies too young to have been vaccinated [6]. A recent report indicated that Hib transmission continues at a low rate in The Gambia without a booster vaccination [34]. Additionally, cases of *Haemophilus influenzae* type a are gradually increasing as it accounts for 83% (5/6) of the non-type b *Haemophilus influenzae*. This observation was also made by previous studies from

both high and low-and middle-income countries [6,35,36]. Our study included several cases of meningitis due to *Klebsiella pneumoniae*, *Escherichia coli*, non-typhoidal Salmonella and other coliforms agreeing with previous studies that have drawn attention to the important role of gram-negative bacilli in ABM [10,37,38].

Most bacteria that cause meningitis in the rural Gambia are still highly susceptible to cephalosporin, and cefotaxime was identified as the most effective against gram-negative bacilli. An urban Gambia Invasive Bacterial Disease report showed a similar antibiotic resistance pattern [28]. In 2012, ciprofloxacin was used as prophylaxis during meningitis outbreak caused by *Neisseria meningitidis* serogroup W(15). Additionally, our study showed that there has been an overall reduction in cases of antimicrobial resistant pneumococcal vaccine serotypes, but resistance is increasing in non-vaccine serotypes following the introduction of pneumococcal conjugate vaccine.

Access to appropriate diagnostics and effort in reducing diagnostic gaps is a significant global challenge. A recent study has described diagnosis as the biggest gap in the cascade of care caused by many factors [39]. The report corroborates findings from our study, underscoring the urgent need for newer molecular diagnostics [40–43] that better identify and distinguish viral, bacterial and fungal pathogens, to guide earlier and more appropriate clinical management of children with meningitis in sub-Saharan Africa. The time is now to employ molecular PCR approaches as routine investigation to detect pathogens from invasive and non-invasive clinical samples from ill children from sub-Saharan Africa. In addition, surveillance against the emergence of antibiotic resistant pathogen needs to be sustained to determine future trends in antibiotic resistance.

This study had several limitations, which included an inability to employ molecular approaches to diagnose the aetiology of meningitis, the absence of viral and fungal diagnoses, a lack of capacity to do detailed follow-up after discharge from hospital to account for either acute or severe neurological sequelae outcome and lack of consideration for assessment of the length of hospitalisation in the analysis.

There is a continuing burden of ABM in low income countries like The Gambia despite the introduction of effective conjugate vaccines. There is a need to support the World Health Organisation recently lunched new meningitis global strategic goals [44] which aims to defeat meningitis by 2030 and to save over 200,000 lives annually. We recommend increased access to rapid diagnostic and PCR tests across the hospital system to help prevent cases of meningitis being missed, improve both overall clinical care and contribute to an improved understanding of the changing aetiology and epidemiology of ABM in a particular area. Interventions to decrease meningitis further should aim to; introduce affordable pentavalent meningococcal conjugate vaccine against *N. meningitidis* serogroups ACWYX and introduce higher valency pneumococcal conjugate vaccines such as PCV15 or PCV20 or PCV24 to provide wider protection against pneumococcal vaccine serotypes. Continuing surveillance will be required for monitoring trends in the evolution of serogroups and serotypes following introduction of new vaccines.

## Acknowledgments

We thank the staff of Basse District Hospital, and staff of other health facilities in Upper River Region. We also thank the staff of the Expanded Programme on Immunisation and the Gambia government for their unwavering collaboration with the Medical Research Council Unit at the London School of Hygiene and Tropical Medicine The Gambia. We thank all staff who worked on the Pneumococcal Surveillance Programme and Basse Health Demographic Surveillance System (BHDSS) for their support and the residents living in the area surveyed by the BHDSS team for their participation in the study.

## Author Contributions

**Conceptualization:** Usman N. Ikumapayi, Grant A. Mackenzie.

**Formal analysis:** Usman N. Ikumapayi, Yekini Olatunji.

**Funding acquisition:** Grant A. Mackenzie.

**Investigation:** Ilias Hossain, Yekini Olatunji, Malick Ndiaye, Henry Badji, Ahmed Manjang, Rasheed Salaudeen, Lamin Ceesay, Richard A. Adegbola, Brian M. Greenwood, Grant A. Mackenzie.

**Resources:** Grant A. Mackenzie.

**Supervision:** Richard A. Adegbola, Brian M. Greenwood, Grant A. Mackenzie.

**Visualization:** Usman N. Ikumapayi.

**Writing – original draft:** Usman N. Ikumapayi.

**Writing – review & editing:** Philip C. Hill, Richard A. Adegbola, Brian M. Greenwood, Grant A. Mackenzie.

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
