## [Decision Letter · Decision Letter 0]

1 Apr 2022

PONE-D-22-05903Childhood meningitis in rural Gambia: 10 years of population-based surveillancePLOS ONE

Dear Dr. Ikumapayi,

Thank you for submitting your manuscript to PLOS ONE. After careful consideration, we feel that it has merit but does not fully meet PLOS ONE’s publication criteria as it currently stands. Therefore, we invite you to submit a revised version of the manuscript that addresses the points raised during the review process.

 Two reviewers agreed that your manuscript would benefit from further changes and made many constructive comments. Please address all of them before resubmitting.

We look forward to receiving your revised manuscript.

Kind regards,

Joël Mossong

Academic Editor

PLOS ONE

Journal Requirements:

Reviewers' comments:

Reviewer's Responses to Questions

**Comments to the Author**

1. Is the manuscript technically sound, and do the data support the conclusions?

Reviewer #1: Yes

Reviewer #2: Yes

Reviewer #3: Partly

2. Has the statistical analysis been performed appropriately and rigorously? 

Reviewer #1: Yes

Reviewer #2: Yes

Reviewer #3: No

3. Have the authors made all data underlying the findings in their manuscript fully available?

Reviewer #1: Yes

Reviewer #2: Yes

Reviewer #3: Yes

4. Is the manuscript presented in an intelligible fashion and written in standard English?

Reviewer #1: Yes

Reviewer #2: Yes

Reviewer #3: Yes

5. Review Comments to the Author

Reviewer #1: In this article, the authors describe the disease burden and microbiological features of acute bacterial meningitis, non-bacterial meningitis, and suspected meningitis in rural Gambia through analysis of population-based surveillance data during 2008-2018. This is an interesting and important paper, but some re-structuring, elaboration, and clarifications are needed to help the reader interpret the results.

Major comments:

1. The presentation of etiology in the results section includes serotype and serogroup results for Neisseria meningitidis and Haemophilus influenzae; however, serogrouping and serotyping of these two pathogens, respectively, are not mentioned in the methods. Please discuss in the methods section.

2. The percentages in Table 1 for age categories are confusing to interpret, as they do not add up to 100%. Additionally, in the text, the primary outcomes of interest are stated as incidence of ABM and NBM vs. CSM, but in Table 1, the groupings don’t reflect this (NBM and CSM are combined into a single category). I would recommend re-grouping the categories in the table to align with the text.

a. In particular, lines 173-174 reference Table 1 to compare gender distribution of ABM and NBM, but table 1 does not have NBM by itself, so the comparison between ABM and NBM can’t be made. I recommend either removing or rephrasing this sentence or re-structuring the table.

3. Figure 1 shows that patients who only had a CSF collected were much more likely to be positive (92/(92+258) = 26%) than those who had both blood and CSF (53/(53+711) = 7%). This suggests that there is something systematically different between patients who had both blood and CSF collected and those who had CSF alone. Can the authors shed light on this point? Were blood samples collected only if they couldn't get a CSF or if the CSF was negative? It would be interesting to see data on the concordance between blood and CSF culture results.

a. In addition, it would be helpful to provide more information in the methods on specimen collection methods/criteria, particularly the volume of specimen collected for each patient and if there were different criteria for collecting blood vs. CSF specimens.

Minor Comments:

1. The authors mention that children with clinical signs of meningitis who had a positive rapid ICT diagnostic test for malaria and normal leukocyte count were excluded from the analysis. It would be interesting to know how many children were excluded from the analysis each year for this reason.

a. In addition, in lines 184-187, an association is made between high meningitis incidence and increased cerebral malaria in 2011. If there was a greater number of patients who tested positive for malaria in 2011 compared to other years, please state this to substantiate the claim of association. Lastly, discussion of this interpretation should be in the discussion section instead of results.

2. There is little information provided on patients presenting with clinical meningitis signs without a sample collected are accounted for in the analysis, although they account for over 10% of patients with clinical signs of meningitis (172/1,599). It would be helpful to discuss barriers that contributed to no samples being collected among these patients, and if possible, the case fatality rate or any follow-up clinical observations made of these patients.

3. Throughout the paper, the authors do not mention the role of seasonality in their analysis. Literature suggests that the Gambia experiences similar meningitis season patterns as seen in other meningitis belt countries. It would be interesting to see how the incidence of ABM, NBM, and CSM differ between the dry and rainy seasons during the 10 years. An epi curve could be helpful to depict this relationship.

4. Line 67: “meningitis-belt” should not be hyphenated.

5. Line 68: the authors say that the region “still” experiences high rates of meningitis. It could be helpful to provide some context – “still” in reference to when? Or the authors can just say a region that has historically experienced high rates…

6. Typo in line 72: “N. meningitides” should be “N. meningitidis”

7. Line 97: sentence starting with “The estimated…,” should be moved to results section.

8. Lines 121-124 “All bacterial antigen…” should be moved to results section.

9. In the second half of table 1 (Died, Survived, Total), the percentages would be more informative if they were presented as row percentages, showing case fatality rate for ABM, CSM, and NBM (instead of breaking down the percentage of deaths into disease categories).

10. In Table 2a, there is a sharp decline in number of cases and incidence reported in 2016 and 2017 for NBM and ABM, and to a lesser extent, CSM. Are there any hypothesized reasons for this?

11. Line 201-202 states CFR “varied significantly” and references Figure 2a, but this figure does not present any statistical testing results. I recommend changing the wording to something along the lines of “varied drastically.”

12. Please clarify whether “Non-Hib” in Table 3 is meant to say, “Hi non-b” and change accordingly.

13. Lines 243-246: It is confusing why certain antibiotics are highlighted in the text and not others. I suggest providing a broader summary statement of the full breadth of testing that was done (e.g. most bacterial isolates were susceptible to clinically relevant antibiotics) and clearly explaining why certain results are referenced in the text (e.g. of antibiotics commonly used for ABM treatment...)

14. Lines 285-287 “Our study also showed…” presents new data analysis findings and is in the discussion section. Please move this to results.

15. The author references cumulative incidence from a study that was done in 2013 (line 309), but this value needs to be put in the context of specific time frame. Was it annual incidence? A specific period? Please specify.

16. Line 326 “against the emergence of antibiotics” – do you mean “antibiotic resistance?”

Reviewer #2: This is a well-written article about the epidemiology of meningitis in Ghana between 2008 and 2017. While authors detected low number of culture positive cases and also high number of suspected meningitis cases due to malaria with neurological involvement, 10 years findings show accurate data for real world vaccine effectiveness.

I have some minor points:

1- Authors need to add vaccine schedule for conjugated pneumococcal vaccine (3 plus one or 3 plus zero or other).

2- Case Fatality rate of pneumococcal meningitis and sepsis (positive blood culture) was very high, 34.1% (higher than Neisseria meningitidis). Authors have data about the case fatality rate for each serotypes (I know that the number cases for each serotypes was low but need to add some information about this high CFR for pneumococci, like underlying disease, age or other risk factors if available)? CFR of non-bacterial meningitis cases (culture and gram stain negative) were also higher than expected, 8.6% of children with pleocystosis dies. It is difficult to say something about exact etiological cause of this cases, (especially children below 2 years old) however it seems bacterial causes like S. pneumoniae or N. meningitidis (my personal opinion).

3- Maybe authors could add some information about newly authorized conjugated pneumococcal vaccine coverage (PCV15 and PCV20) for this settings, especially post PCV13 era.

Reviewer #3: General comments

The manuscript presents important epidemiological data on the incidence and case fatality rates for different clinical categories of meningitis (CSM, ABM, NBM [see below]) among rural Gambian children for an extended period of time (2008-2017).

Critically, the data presented indicates case fatality rates for meningitis have not declined over time (Figure 2a). Similarly, the overall incidence for meningitis does not appear to have declined over time - Table 2a (despite vaccine strategies). Culture proven bacterial meningitis again appear to represent a low number of suspected cases.

The manuscript highlights a gap in diagnosis, with continued high incidence of clinically suspected meningitis (and sub-groups), with child cases continuing to demonstrate poor outcome (particularly among ABM cases).

The manuscript would benefit from some clarification of the methods and definitions employed. It would also benefit from more detailed documentation and employment of statistics to further support observations seen in the Tables/Graphs.

Clinical Categories analysed:-

CSM – clinically suspected meningitis

ABM – acute bacterial meningitis

NBM – non-bacterial meningitis

In addition, the authors describe the current diagnostic pathway for The Gambia/Upper River Region of The Gambia. They highlight the need for “newer molecular diagnostics”, indicating some level of molecular testing is available (despite not being presented in this study).

Specific comments

Authors state (line 117) - “The definition of acute bacterial meningitis (ABM) was a CSF leucocyte count >5/mm3, and a positive culture of CSF or blood in a patient with clinical signs of meningitis”. However, they also state (line 170) “Blood and CSF were obtained from 764 children, blood only from 306, and CSF only from 171”.

I understand from Figure 1 and ‘surveillance procedures’ that of the 1,427 samples cultures included in this study, 24/169 positive cultures, 34/41 contaminant and 248/1,217 Negative cultures, were derived from blood only samples (n=306). Is this correct? It would be useful to add the three clinical categories (CSM, ABM and NBM) to Figure 1 to clarify what sampling was performed for each category. Further methodological description is needed to confirm that whilst microbiological investigations were carried out on blood only sampling (n=306), ASM and NBM categories were confirmed via sampling of both blood and CSF, with a positive or negative culture in CSF and/or blood needed to fit either category, respectively.

It would be useful to present the range for CSF white cell count among all ABM cases (where LP performed), particularly comparing CSF leukocyte count among ABM cases identified by positive CSF or blood culture.

Among those cases where blood only was collected, how did authors distinguish between blood stream bacterial infection (positive blood culture) with impaired consciousness and acute bacterial meningitis (ABM)?

ABM definition used in this manuscript doesn’t typically apply to infants under 1 month (CSF >5 cells/mm3 is defined as a normal among children <1 month of age). For infants under 1 month of age, higher levels (e.g. 10-22 cells/mm3) are accepted as a cut off for abnormal. What portion of ABM cases were aged under 1 month. Application of the CSF leukocyte cut-off <5cells/mm3 to children <1 month may potentially contribute to the manuscript’s high estimated incidence of NBM among children under 2months of age. Incorporating this cut off within the current ABM definition and reanalysing the category assignment for this age group is needed.

How did the authors define children with symptoms of meningitis, positive CSF bacterial culture and CSF leukocytes below 5 cells/mm3? Were these cases defined as “contaminants” (documented in Figure 1)? It would be good to know what pathogens were detected in the ‘contaminant’ group. As the authors are aware, you can have meningitis (with CSF positive bacterial culture) without a raised CSF leukocyte count, particularly among immuno-supressed children.

NBM is potentially a misleading term. I suggest renaming the “NBM – non-bacterial meningitis” clinical category to “SNBM – Suspected non-bacterial meningitis” or “NCM – non-categorised meningitis”, since no diagnostics were employed to identify viral or fungal pathogen, or exclude bacterial meningitis, whilst negative by culture, that may have been detected by molecular methods.

Laboratory methods

Authors do not provide information on which bacterial species were investigated using bacterial antigen tests. They do not provide information on which tests/kit were employed. Gold standard diagnosis is typically via culture (not antigen tests). It would be useful to present which ABM cases were identified using antigen tests only (i.e. culture negative) and the correspondence between culture positive and positive antigen tests for CSF or blood samples.

Whilst CSF bacterial antigen test is not the gold standard, authors note CSF antigen were paired with CSF culture, and tested according to WHO procedures. They also indicate 100% accuracy of antigen tests compared with culture in CSF (lines 121-124). It would be useful for authors to additionally describe the common practise for laboratory diagnosis in the region within the ‘surveillance procedures’ (e.g. whether kits are employed for presumptive diagnosis whilst waiting for culture or confirmatory following culture or gram stain).

Statistics

Authors should define how p values, odds ratios and confidence intervals were derived and what software package was employed. Authors should also define and present further detail on how disease incidence was calculated.

Results

Table 1 Baseline Characteristics

Unclear how the percentages (%) are derived in age sub-groups across the three clinical categories.

I suggest the denominator should be all ages (within each clinical category).

For example, in the sub-group <2 months within ABM

<2mo n=27; all ages n=169 – therefore % aged 2mo is 27/169=16% (not 50% as presented)

Odds ratios need to be presented with associated P values

Similarly, for Outcome – denominator should be all cases (doesn’t make clinical sense to pick NBM as the reference group)

Odds ratio should be associated with P values again

So for

ABM died n=49; Survived=120; % died 49/169= 28.99%; P value <0.001; odds ratio 2.97

NBM died n=18; Survived=191; % died 18/209 = 8.6%; P val. = 0.019; OR 0.53

CSM died n=134; survived n=915; % died = 134/1049 = 12.77; P val.=0.022; OR 0.68

Overall (all groups) died n=201; survived n=1226; % died 201/1427; = 14.1%

i.e. ABM significant higher proportion of deaths in ABM group compared to proportion in all groups combined.

and NBM and CSM significantly lower proportion of deaths compared to all groups combined (lowest among NBM)

Table 3: It would be useful to present associated P values for each pathogen that shows a statistically higher number of cases associated with death compared to all pathogens across all ABM cases (or confirm no statistical difference for any individual pathogen).

Table 4: It would be useful to present associated P values for Chi square test of vaccine associated vs non-vaccine associated serotypes pre and post PCV13 vaccine campaign.

Table 5: Given the authors have extensive antibiotic resistance on bacteria from 2008-2017, the manuscript would be strengthened by examining this data for changes in antibiotic resistance patterns over time (at least for the more commonly identified pathogens [e.g. S.Pneu; H.Infl; N.Men]). Publications from other regions and other sub-Saharan countries (e.g. Malawi) are reporting increased antibiotic resistance rates over time.

Discussion

Line 263. “Over the study period, there was steady decrease in the incidence of ABM except in 2012...” This statement does not appear to reflect the results presented in Table 2a on ABM cases and estimated incidence between 2008-2017. Authors should ideally present statistical analyses in results to support their statement. One could potentially argue decline is only seen in 2016-2017.

Authors briefly comment on the potential for missed diagnosis in the NBM category with prior-antibiotic use before LP (Line 278). Would be good to expand on the guideline’s vs practise for LP, and usage of empirical antibiotics in the region. Line 326 is incomplete/unclear: “emergence of antibiotics”, do you mean emergence of additional antibiotics or emergence of antibiotic resistance?

Overall, the language used in the manuscript is clear, although there are minor typographical errors the authors should correct during revision.

6. PLOS authors have the option to publish the peer review history of their article (what does this mean?). If published, this will include your full peer review and any attached files.

Reviewer #1: No

Reviewer #2: No

Reviewer #3: No

---

## [Author Response · Author response to Decision Letter 0]

26 Jun 2022

Please find below from page 2 to page 23 our responses to the reviewers’ comments. All the comment made by the reviewers were very helpful, we have made utmost efforts to respond to each of the comment clearly and succinctly. The responses were in font purple colour plus responses included in the manuscript or text were indicated in italics with quotation. 

Additionally, the correct funding disclosure statement is “This work was supported by the Bill & Melinda Gates Foundation (grant number OPP1020327) and the Medical Research Council Unit The Gambia at London School of Hygiene and Tropical Medicine.” The grant recipient was Dr Grant A Mackenzie.

following response to reviewers’ comment. 

Reviewers' comments:

Reviewer's Responses to Questions

Comments to the Author

1. Is the manuscript technically sound, and do the data support the conclusions?

Reviewer #1: Yes

Reviewer #2: Yes

Reviewer #3: Partly

2. Has the statistical analysis been performed appropriately and rigorously? 

Reviewer #1: Yes

Reviewer #2: Yes

Reviewer #3: No

3. Have the authors made all data underlying the findings in their manuscript fully available?

Reviewer #1: Yes

Reviewer #2: Yes

Reviewer #3: Yes

4. Is the manuscript presented in an intelligible fashion and written in standard English?

Reviewer #1: Yes

Reviewer #2: Yes

Reviewer #3: Yes

5. Review Comments to the Author

Reviewer #1: 

In this article, the authors describe the disease burden and microbiological features of acute bacterial meningitis, non-bacterial meningitis, and suspected meningitis in rural Gambia through analysis of population-based surveillance data during 2008-2018. This is an interesting and important paper, but some re-structuring, elaboration, and clarifications are needed to help the reader interpret the results.

Major comments:

1. The presentation of etiology in the results section includes serotype and serogroup results for Neisseria meningitidis and Haemophilus influenzae; however, serogrouping and serotyping of these two pathogens, respectively, are not mentioned in the methods. Please discuss in the methods section.

The methods used for serogrouping and serotyping sentences have now been added to the methods section as requested (lines 171-176)

‘Neisseria meningitidis isolates’ serogroup were identified by latex agglutination using Ramel (Thermo Fisher Scientific, Waltman, MA, USA) test kits. Haemophilus influenzae type b and other encapsulated H. influenzae other than type b isolates were distinguished by slide latex agglutination using polyvalent and monovalent Haemophilus influenzae Difco serotyping antisera to Hi of types a, b, c, d, e and f (Beckton Dickinson, Difco, Belgium)’

2. The percentages in Table 1 for age categories are confusing to interpret, as they do not add up to 100%.

We apologise for this mistake which has now been corrected in table 1 (line 251-253)

Additionally, in the text, the primary outcomes of interest are stated as incidence of ABM and NBM vs. CSM, but in Table 1, the groupings don’t reflect this (NBM and CSM are combined into a single category). I would recommend re-grouping the categories in the table to align with the text. 

Thank you for this helpful observation, that section of Table 1 has been separated and now named Table 3 and amendment made to match the text (line298-300). 

a. In particular, lines 173-174 reference Table 1 to compare gender distribution of ABM and NBM, but table 1 does not have NBM by itself, so the comparison between ABM and NBM can’t be made. I recommend either removing or rephrasing this sentence or re-structuring the table. 

Table 1 has been re-structured as suggested (line251-253)

3. Figure 1 shows that patients who only had a CSF collected were much more likely to be positive (92/(92+258) = 26%) than those who had both blood and CSF (53/(53+711) = 7%). This suggests that there is something systematically different between patients who had both blood and CSF collected and those who had CSF alone. Can the authors shed light on this point? Were blood samples collected only if they couldn't get a CSF or if the CSF was negative? 

According to the study protocol it is mandatory to collect blood and CSF from recruited children with clinical presentation of meningitis unless the patient is very sick and severely dehydrated that made it impossible for blood to be collected. In this situation only CSF is collected which explains higher isolation rate in the CSF alone group. 

It would be interesting to see data on the concordance between blood and CSF culture results. 

Our data showed 100% concordance.

a. In addition, it would be helpful to provide more information in the methods on specimen collection methods/criteria, particularly the volume of specimen collected for each patient and if there were different criteria for collecting blood vs. CSF specimens. 

Thanks for this observation, we have now added this information to methods section (line 151-152).

‘A 3 to 5 ml venous blood sample was collected from study participants. Lumbar puncture was performed to obtain 1 to 2 ml CSF’.

Minor Comments:

1. The authors mention that children with clinical signs of meningitis who had a positive rapid ICT diagnostic test for malaria and normal leukocyte count were excluded from the analysis. It would be interesting to know how many children were excluded from the analysis each year for this reason.

On average, 5 children were excluded yearly. However, 67 children were excluded throughout the study period and this has been added to the figure 1 and text under method (line 181) ‘Sixty-seven’. 

a. In addition, in lines 184-187, an association is made between high meningitis incidence and increased cerebral malaria in 2011. If there was a greater number of patients who tested positive for malaria in 2011 compared to other years, please state this to substantiate the claim of association. 

This is genuine observation. Our expression was not meant to be categorical but hypothetical. So, sentence has been rephrased to (line 465-467) 

‘Over the study period, there was no considerable decrease in the incidence of ABM except in 2016 and 2017 which may be due to low rainfalls resulted in low malaria transmission with subsequent declined cerebral malaria presentation.’

Lastly, discussion of this interpretation should be in the discussion section instead of results. 

This has been moved to discussion section (line 465-467) as suggested.

‘Over the study period, there was no considerable decrease in the incidence of ABM except in 2016 and 2017 which may be due to low rainfalls resulted in low malaria transmission with subsequent declined cerebral malaria presentation.’

2. There is little information provided on patients presenting with clinical meningitis signs without a sample collected are accounted for in the analysis, although they account for over 10% of patients with clinical signs of meningitis (172/1,599). It would be helpful to discuss barriers that contributed to no samples being collected among these patients, and if possible, the case fatality rate or any follow-up clinical observations made of these patients. 

Thank you for raising this important point. Reasons why CSF and blood culture samples were not obtained from these patients have now been included in the text (line 216-223) and case fatality rates have been included in the results section (lines 216 and 223).

‘Of those with clinical signs of meningitis cases, 14.3% (239/1666) were excluded from analysis. Of these, 72.0% (172/239) were due to blood and CSF not being taken for culture as the children were severely ill and dehydrated, 7.1 % of these cases (17) died and 9.6% (23) were referred prior to sample collection. The other 28% (67/239) were cases of clinical sign meningitis that were positive rapid ICT test for malaria, and there were 7.4% (5) deaths from this group’.

3. Throughout the paper, the authors do not mention the role of seasonality in their analysis. Literature suggests that the Gambia experiences similar meningitis season patterns as seen in other meningitis belt countries. It would be interesting to see how the incidence of ABM, NBM, and CSM differ between the dry and rainy seasons during the 10 years. An epi curve could be helpful to depict this relationship. 

Thank you for this helpful suggestion. A figure showing the relation between season and cases of meningitis has now been included and indicated as Figure 3. 

‘Figure 3: Seasonal Distribution of Clinically Suspected Meningitis (CSM), Suspected Non-Bacterial Meningitis (SNBM) and Acute Bacterial Meningitis (ABM) Over 10 Years in Upper River Region Gambia, 2008-2017’

This figure has now been reported in the results section (line 301-303)

‘Seasonality

The three categories of clinical signs of meningitis were proportionately higher during the dry season (November-June) compared to the wet season (July-October) (Figure 3)’.

Similarly, seasonality has now been discussed in the discussion section (line 553-556)

‘The ABM, SNBM and CSM were consistently higher in the dry season compared to the wet season. ABM was higher compared to SNBM and CSM in dry season, but in the wet season CSM was higher compared to ABM and CSBM. This observation is consistent with other studies from sub-Saharan Africa associating spread of meningitis with dry weather’

4. Line 67: “meningitis-belt” should not be hyphenated. 

This change has been made (line 74-75) ‘meningitis belt’

5. Line 68: the authors say that the region “still” experiences high rates of meningitis. It could be helpful to provide some context – “still” in reference to when? Or the authors can just say a region that has historically experienced high rates… 

The latter suggestion of the reviewer has been accepted (line 75) ‘has historically experienced’ 

6. Typo in line 72: “N. meningitides” should be “N. meningitidis” 

This has been corrected (line 79) ‘N. meningitidis’

7. Line 97: sentence starting with “The estimated…,” should be moved to results section. 

This is useful observation and we too thought that the sentence should not be where it is but we later realized that we are only trying to provide information about the study site. So we decided to maintain it under the method section (now line 115) ‘The estimated’

8. Lines 121-124 “All bacterial antigen…” should be moved to results section. 

This change has been made (line 227-230) 

‘All bacterial antigen positive CSF were also culture positive although some CSF had very scanty growth of <20 x 105 organisms/mL. All bacterial antigen negative CSF were culture negative. However, only 87% of cultured CSF samples were tested for bacterial antigens’.

9. In the second half of table 1 (Died, Survived, Total), the percentages would be more informative if they were presented as row percentages, showing case fatality rate for ABM, CSM, and NBM (instead of breaking down the percentage of deaths into disease categories). 

We accept that this would be a useful way of presenting these data but would prefer to retain the current format because it is important to provide strong evidence of the clinical importance of ABM using statistical evaluation to report odd ratio, 95% Confidence interval and P-value it is now Table 3 (line 298-300)

 ‘Table 3: Fatal outcome of meningitis in rural Gambia: 10 years of population-based surveillance’

10. In Table 2a, there is a sharp decline in number of cases and incidence reported in 2016 and 2017 for NBM and ABM, and to a lesser extent, CSM. Are there any hypothesized reasons for this? 

A sentence has been added to the discussion covering this point (line 465-467) 

‘no considerable decrease in the incidence of ABM except in 2016 and 2017 which may be due to low rainfalls resulted in low malaria transmission with subsequent declined cerebral malaria presentation’.

11. Line 201-202 states CFR “varied significantly” and references Figure 2a, but this figure does not present any statistical testing results. I recommend changing the wording to something along the lines of “varied drastically.” 

This change has been made (line 288) the word significantly changed to ‘considerably’

12. Please clarify whether “Non-Hib” in Table 3 is meant to say, “Hi non-b” and change accordingly.

This has been changed to non-type b H. influenzae as suggested throughout the text

13. Lines 243-246: It is confusing why certain antibiotics are highlighted in the text and not others. I suggest providing a broader summary statement of the full breadth of testing that was done (e.g. most bacterial isolates were susceptible to clinically relevant antibiotics) and clearly explaining why certain results are referenced in the text (e.g. of antibiotics commonly used for ABM treatment...) 

Thanks for the useful observation. All clinically relevant antibiotics against ABM have now been mentioned (line 457-458) also highlight removed

14. Lines 285-287 “Our study also showed…” presents new data analysis findings and is in the discussion section. Please move this to results. 

This has been done as suggested and this sentence is now in the results (line 294-297). 

‘Our study also showed that 100% (49/49) of the ABM related death occurred within the first 5 days of hospitalization. Whilst SNBM and CSM related death occurred within 3 and 15 days respectively. The number of deaths following hospital discharge for SNBM and CSM were 2/18 and 9/134 respectively’.

15. The author references cumulative incidence from a study that was done in 2013 (line 309), but this value needs to be put in the context of specific time frame. Was it annual incidence? A specific period? Please specify. 

This has been clarified and an additional sentence added (line 584-585) ‘from 2010 to 2013’

16. Line 326 “against the emergence of antibiotics” – do you mean “antibiotic resistance?”

This change has been made with additional sentences (line 623-624) 

‘surveillance against the emergence of antibiotic resistant pathogen needs to be sustained to determine future trends in antibiotic resistance’

Reviewer #2: 

This is a well-written article about the epidemiology of meningitis in Ghana between 2008 and 2017. While authors detected low number of culture positive cases and also high number of suspected meningitis cases due to malaria with neurological involvement, 10 years findings show accurate data for real world vaccine effectiveness.

I have some minor points:

1- Authors need to add vaccine schedule for conjugated pneumococcal vaccine (3 plus one or 3 plus zero or other). 

This information is now provided in the text (line 464-465) ‘3 plus 0 schedules

2- Case Fatality rate of pneumococcal meningitis and sepsis (positive blood culture) was very high, 34.1% (higher than Neisseria meningitidis). Authors have data about the case fatality rate for each serotypes (I know that the number cases for each serotypes was low but need to add some information about this high CFR for pneumococci, like underlying disease, age or other risk factors if available)? 

The mortality of children with pneumococcal meningitis was very high as has been noted in many previous studies in sub-Saharan Africa including previous studies in The Gambia. 

The point raised has been added to the discussion (line 564-577) 

‘The only pneumococcal meningitis death by vaccine serotype in post-PCV13 vaccination was aged 24 months caused by serotype 23F in August 2013, although the child may have died from immunocompromised complications because the mother was HIV tested positive after the death of the child. The two pneumococcal meningitis deaths in non-vaccine serotypes (serotype 2 and 18A) in post-PCV13 were of ages 7days and 28 days in July and November 2017 respectively, and one pneumococcal meningitis death aged 36 months caused by serotype 25F in February 2014 emphasised the need to introduce high valency PCV into the country’s Extended Immunization Program (EPI)’.

CFR of non-bacterial meningitis cases (culture and gram stain negative) were also higher than expected, 8.6% of children with pleocystosis dies. It is difficult to say something about exact etiological cause of this cases, (especially children below 2 years old) however it seems bacterial causes like S. pneumoniae or N. meningitidis (my personal opinion). 

This point has been added to the text (line 615-617)

‘Access to appropriate diagnostics and effort in reducing diagnostic gaps is a significant global challenge. A recent study has described diagnosis as the biggest gap in the cascade of care caused by many factors’

3- Maybe authors could add some information about newly authorized conjugated pneumococcal vaccine coverage (PCV15 and PCV20) for this settings, especially post PCV13 era. 

A sentence has been added to the discussion in response to this suggestion. (line 647-652)

‘Interventions to decrease meningitis further should aim to; introduce affordable pentavalent meningococcal conjugate vaccine against N. meningitidis serogroups ACWYX and introduce higher valency pneumococcal conjugate vaccines such as PCV15 or PCV20 or PCV24 to provide wider protection against pneumococcal vaccine serotypes’. Continuing surveillance will be required for monitoring trends in evolution of serogroups and serotypes following introduction of new vaccines.’ 

Reviewer #3: General comments

The manuscript presents important epidemiological data on the incidence and case fatality rates for different clinical categories of meningitis (CSM, ABM, NBM [see below]) among rural Gambian children for an extended period of time (2008-2017).

Critically, the data presented indicates case fatality rates for meningitis have not declined over time (Figure 2a). Similarly, the overall incidence for meningitis does not appear to have declined over time - Table 2a (despite vaccine strategies). Culture proven bacterial meningitis again appear to represent a low number of suspected cases.

The manuscript highlights a gap in diagnosis, with continued high incidence of clinically suspected meningitis (and sub-groups), with child cases continuing to demonstrate poor outcome (particularly among ABM cases).

The manuscript would benefit from some clarification of the methods and definitions employed. It would also benefit from more detailed documentation and employment of statistics to further support observations seen in the Tables/Graphs.

Clinical Categories analysed:-

CSM – clinically suspected meningitis

ABM – acute bacterial meningitis

NBM – non-bacterial meningitis

In addition, the authors describe the current diagnostic pathway for The Gambia/Upper River Region of The Gambia. They highlight the need for “newer molecular diagnostics”, indicating some level of molecular testing is available (despite not being presented in this study).

Specific comments

Authors state (line 117) - “The definition of acute bacterial meningitis (ABM) was a CSF leucocyte count >5/mm3, and a positive culture of CSF or blood in a patient with clinical signs of meningitis”. However, they also state (line 170) “Blood and CSF were obtained from 764 children, blood only from 306, and CSF only from 171”.

The reviewer has interpreted the definition of acute bacterial meningitis correctly - the child had to have evidence of CSF inflammation and a positive CSF or blood culture. It is possible that a bacterial diagnosis might have been missed in children who had evidence of meningeal inflammation, a negative CSF culture but in whom blood culture was not done but this number have been very small.

I understand from Figure 1 and ‘surveillance procedures’ that of the 1,427 samples cultures included in this study, 24/169 positive cultures, 34/41 contaminant and 248/1,217 Negative cultures, were derived from blood only samples (n=306). Is this correct? 

Yes, this is correct that 24/169 positive cultures, 34/41 contaminant and 248/1,217 Negative cultures, were derived from blood only samples (n=306).

It would be useful to add the three clinical categories (CSM, ABM and NBM) to Figure 1 to clarify what sampling was performed for each category. Further methodological description is needed to confirm that whilst microbiological investigations were carried out on blood only sampling (n=306), ASM and NBM categories were confirmed via sampling of both blood and CSF, with a positive or negative culture in CSF and/or blood needed to fit either category, respectively. 

Thanks for this thoughtful suggestion. The three clinical categories (ABM, CSM and SNBM) have now been added to the figure 1. 

It would be useful to present the range for CSF white cell count among all ABM cases (where LP performed), particularly comparing CSF leukocyte count among ABM cases identified by positive CSF or blood culture. 

This is another thoughtful suggestion which has now been added in the result section (line 248-250) 

‘There were 14 children under 2 months of age who are in the ABM category and had CSF culture and cell count performed, and all had CSF leukocyte counts >10 cells/mm3 (range 45 to 1240cells/mm3)’

Among those cases where blood only was collected, how did authors distinguish between blood stream bacterial infection (positive blood culture) with impaired consciousness and acute bacterial meningitis (ABM)? 

This is a genuine observation. According to the main study (Pneumococcal Surveillance Program) protocol, blood is the primary clinical sample to obtain from a child presented with clinical symptom that has the potential implication of pneumococcal infection. Blood will not be taken only when the child is too ill and severely dehydrated. Thus, a child with a clinical diagnosis but positive blood culture was obtained often had a raised CSF count making them a case of ABM. Additionally, the definition of clinical signs of meningitis that include stiff neck and reduced consciousness suggestive of clinical signs of meningitis. 

ABM definition used in this manuscript doesn’t typically apply to infants under 1 month (CSF >5 cells/mm3 is defined as a normal among children <1 month of age). For infants under 1 month of age, higher levels (e.g. 10-22 cells/mm3) are accepted as a cut off for abnormal. What portion of ABM cases were aged under 1 month. Application of the CSF leukocyte cut-off <5cells/mm3 to children <1 month may potentially contribute to the manuscript’s high estimated incidence of NBM among children under 2months of age. Incorporating this cut off within the current ABM definition and reanalysing the category assignment for this age group is needed. 

There were only 25 children under one month of age in the study. Of the twenty-five, only two fall in the ABM category and they both had a CSF WBC count above 49 cells per mm3 so we do not consider a reanalysis necessary.

 However, we have mentioned about the range of WBC count for children under 2-month of age in the text (line248-250).

‘There were 14 children under 2 months of age who were in the ABM category and had CSF culture and cell count performed, and all had CSF leukocyte counts >10 cells/mm3.(range 45 to 1240cells/mm3)’.

How did the authors define children with symptoms of meningitis, positive CSF bacterial culture and CSF leukocytes below 5 cells/mm3? Were these cases defined as “contaminants” (documented in Figure 1)? It would be good to know what pathogens were detected in the ‘contaminant’ group. 

There were no positive CSF pathogenic bacterial culture with CSF leukocytes count below 5 cells/mm3 that were considered ABM in this study. Likewise, there were no contaminant with CSF leukocytes count above 5 cells/mm3 thus contaminants fall under CSM category and has now been indicated in figure 1. 

The type of environmental contaminant bacterial detected have now been added to the text in the method section (line 233-235) 

‘The organisms considered contaminants in the CSF were Streptococcus viridians 5/7 and coagulase negative Staphylococcus (CNS) 2/7. In the blood, contaminants were Streptococcus viridans 8/34, Micrococcus 6/34, CNS 12/34 and Bacillus species 8/34’.

As the authors are aware, you can have meningitis (with CSF positive bacterial culture) without a raised CSF leukocyte count, particularly among immuno-supressed children. 

Yes, a clinically diagnosed child or adult can have meningitis (with CSF positive bacterial culture) without a raised CSF leukocyte count, particularly among immuno-supressed children. Only one child is suspected of been immuno-suppressed in our study as the mother was HIV tested positive following the death of the child. Detail of this has now been mentioned in discussion section of the text (line 564-567) 

‘The only pneumococcal meningitis death by vaccine serotype in post-PCV13 vaccination was age 24-month caused by serotype 23F in August 2013, although the child may have died from immunocompromised complications because the mother was HIV tested positive after the death of the child’.

Suffice to know that the child’s CSF leukocyte count was 5,120 cells/mm3 

NBM is potentially a misleading term. I suggest renaming the “NBM – non-bacterial meningitis” clinical category to “SNBM – Suspected non-bacterial meningitis” or “NCM – non-categorised meningitis”, since no diagnostics were employed to identify viral or fungal pathogen, or exclude bacterial meningitis, whilst negative by culture, that may have been detected by molecular methods. 

Non-bacterial meningitis is a term widely used to describe this group of patients but we will follow the reviewers suggestion and change this to SNBM – Suspected Non-Bacterial Meningitis throughout the text and figures 

Laboratory methods

Authors do not provide information on which bacterial species were investigated using bacterial antigen tests. They do not provide information on which tests/kit were employed. Gold standard diagnosis is typically via culture (not antigen tests). 

This information is now provided in the method section in the text (line 169-176) 

‘serotyping for pneumococcal serotypes with a latex agglutination assay using factor and group-specific antisera (Statens Serum Institut, Copenhangen, Denmark). Neisseria meningitidis isolates were tested for serogrouping by latex agglutination using Ramel (Thermo Fisher Scientific, Waltman, MA, USA) test kits. H. influenzae type b and other encapsulated H. influenzae other than type b was by slide latex agglutination using polyvalent and monovalent Haemophilus influenzae Difco serotyping antisera to Hi of types a, b, c, d, e and f (Beckton Dickinson, Difco, Belgium)’.

It would be useful to present which ABM cases were identified using antigen tests only (i.e. culture negative) and the correspondence between culture positive and positive antigen tests for CSF or blood samples. 

Again, interestingly, all antigen tests that were tested positive for either Streptococcus pneumoniae, Haemophilus influenzae and Neisseria meningitidis were also culture positive. The only difference was in turn-around-time in which some H. influenzae and N. meningitidis grew very scantily (<20 x 105 organisms/ml after 48hours. This has now been mentioned in the result section (line 227-230)

‘All bacterial antigen positive CSF samples were also culture positive although some CSF had very scanty growth of <20 x 105 organisms/mL after 48 hours. All bacterial antigen negative CSF were culture negative. However, only 87% of cultured CSF samples were tested for bacterial antigens’.

Whilst CSF bacterial antigen test is not the gold standard, authors note CSF antigen were paired with CSF culture, and tested according to WHO procedures. They also indicate 100% accuracy of antigen tests compared with culture in CSF (lines 121-124). It would be useful for authors to additionally describe the common practise for laboratory diagnosis in the region within the ‘surveillance procedures’ (e.g. whether kits are employed for presumptive diagnosis whilst waiting for culture or confirmatory following culture or gram stain). Routinely using 

This aspect has now been added in the method section of the text (Line 176-179)

‘In line with the surveillance procedure and WHO guidelines (WHO/CDS/CSR/EDC/99.7) presumptive diagnosis that include; cell count, Gram-stain and bacterial antigen (latex agglutination) tests were routinely performed on CSF whilst waiting for culture confirmatory result’.

Statistics

Authors should define how p values, odds ratios and confidence intervals were derived and what software package was employed. Authors should also define and present further detail on how disease incidence was calculated. These concerns have already been dealt with in the method section of statistics

We believe that the way incidence rates were calculated is already well described in the statistical section of the minutes. We have added the name of the statistical package used for many of the statistical calculations. 

The statistical package has now been added in the method section (line 199-200) 

‘were done using Stata 14 software package’.

Results

Table 1 Baseline Characteristics

Unclear how the percentages (%) are derived in age sub-groups across the three clinical categories.

I suggest the denominator should be all ages (within each clinical category).

For example, in the sub-group <2 months within ABM

<2mo n=27; all ages n=169 – therefore % aged 2mo is 27/169=16% (not 50% as presented) 

This is a helpful observation. We have now calculated it in the standard format in the current Table1 (line 251-253)

‘Table 1: Baseline meningitis in rural Gambia: 10 years of population-based surveillance’

Odds ratios need to be presented with associated P values 

This observation has now been considered in the current Table 3 (line298-300).

‘Table 3: Fatal outcome of meningitis in rural Gambia: 10 years of population-based surveillance’

Similarly, for Outcome – denominator should be all cases (doesn’t make clinical sense to pick NBM as the reference group) Odds ratio should be associated with P values again 

So for

ABM died n=49; Survived=120; % died 49/169= 28.99%; P value <0.001; odds ratio 2.97

NBM died n=18; Survived=191; % died 18/209 = 8.6%; P val. = 0.019; OR 0.53

CSM died n=134; survived n=915; % died = 134/1049 = 12.77; P val.=0.022; OR 0.68

Overall (all groups) died n=201; survived n=1226; % died 201/1427; = 14.1%

i.e. ABM significant higher proportion of deaths in ABM group compared to proportion in all groups combined and NBM and CSM significantly lower proportion of deaths compared to all groups combined (lowest among NBM) 

The required statistical format has been followed to obtained the adequate odd ratio, 95% Confidence interval and P-value. Please see the current Table 3 (line 298-300)

‘Table 3: Fatal outcome meningitis in rural Gambia: 10 years of population-based surveillance’

Table 3: It would be useful to present associated P values for each pathogen that shows a statistically higher number of cases associated with death compared to all pathogens across all ABM cases (or confirm no statistical difference for any individual pathogen). 

The Table 3 in question has now been renamed Table 4. (line 334-338)

We acknowledged that it may have been informative to show a statistical higher number of each cases of pathogen associated with death. However, we believe case fatality ratio and 95% confidence interval have shown clear representation of each pathogen associated with death. To avoid complication, we prefer to maintain this format. 

Table 4: It would be useful to present associated P values for Chi square test of vaccine associated vs non-vaccine associated serotypes pre and post PCV13 vaccine campaign. 

The table 4 previously has now been renamed Table 5 (line 387-435) 

Yes, we also thought of this useful suggestion, but we were compelled to limit our calculation to proportion because of the limited number of pneumococcal serotypes in the study that does not give us adequate denominator to make comparisons in the incidence. Moreover, the number of cases of pneumococcal vaccine serotype fell from 18 prior introduction of PCV13 to 4 following introduction of PCV13. However, the comparable numbers for non-vaccine serotype pneumococcal meningitis in both Pre and Post PCV13 were 11 and 11 each. Clearly, no informative statistical analysis between the 4 groups can be derived.

Table 5: Given the authors have extensive antibiotic resistance on bacteria from 2008-2017, the manuscript would be strengthened by examining this data for changes in antibiotic resistance patterns over time (at least for the more commonly identified pathogens [e.g. S.Pneu; H.Infl; N.Men]). Publications from other regions and other sub-Saharan countries (e.g. Malawi) are reporting increased antibiotic resistance rates over time. 

The Table 5 in question has now been renamed Table 6 (line 447-460)

Yes, our observation too was that there are increase antibiotic resistance rates over time around different geographical location globally. We have also observed this resistance against few frequently prescribed antibiotics such as cotrimoxazole, oxacillin and tetracycline in our setting. 

However, we require substantial amount of data to address this systematically as requested and that would probably lead to a different study entirely. 

We have tried to comply to the suggestion by revisiting our database and have mentioned our observation in the discussion section in the text (line600-614)

‘Additionally, our study showed that there is overall reduction in cases of antimicrobial resistant pneumococcal vaccine serotypes, but resistance is increasing in non-vaccine serotypes following the introduction of PCV’.

Discussion

Line 263. “Over the study period, there was steady decrease in the incidence of ABM except in 2012...” This statement does not appear to reflect the results presented in Table 2a on ABM cases and estimated incidence between 2008-2017. Authors should ideally present statistical analyses in results to support their statement. One could potentially argue decline is only seen in 2016-2017. 

Thanks for the observation. The misconception has been corrected by rephrasing the sentence (line 465-467) 

‘Over the study period, there was no considerable decrease in the incidence of ABM except in 2016 and 2017 which may be due to low rainfalls resulted in low malaria transmission with subsequent declined cerebral malaria presentation’.

Authors briefly comment on the potential for missed diagnosis in the NBM category with prior-antibiotic use before LP (Line 278). Would be good to expand on the guideline’s vs practise for LP, and usage of empirical antibiotics in the region.

Yes, prior antibiotic use might have been the cause of some failures to obtain a culture positive in cases of SNBM. However, we have now explained in detail in the discussion section in the text (line 522-525)

‘Although the common practice in our formal health system is to obtain LP prior antibiotic usage but antibiotics are available in the study area outside the formal health system. Thus, few cases might have taken antibiotic before presenting to the hospital’.

Line 326 is incomplete/unclear: “emergence of antibiotics”, do you mean emergence of additional antibiotics or emergence of antibiotic resistance? 

This has been clarified in the revised text line (623-624) 

‘ the emergence of antibiotic resistant-pathogen needs to be sustained to determine future trends in antibiotic resistance’.

Overall, the language used in the manuscript is clear, although there are minor typographical errors the authors should correct during revision.

Thanks, this is another important observation. The revised text has been checked for typographical errors.

Reviewer #4: General comment

1. Recommendation (Answer options: Accept, Minor Revision, Major Revision, Reject) 2. 

2. Is the manuscript technically sound, and do the data support the conclusions? (Answer options: Yes, No, Partly) 

3. Has the statistical analysis been performed appropriately and rigorously? (Answer options: Yes, No, I don’t know, N/A) 

4. Does the manuscript adhere to the PLOS Data Policy? Additional details can be found at http://journals.plos.org/plosone/s/materials-and-software-sharing. (Answer options: Yes, No) 

5. Is the manuscript presented in an intelligible fashion and written in standard English? (Answer options: Yes, No) 

6. Review Comments to the Author (minimum 200 characters) 

This study summarises a good quality population-based surveillance for bacterial meningitis 

In the title and abstract to define that this study is among children <15 years. This only becomes clearer in the methods. Were neonates excluded? 

This a very thoughtful question. This was not considered in our initial analysis; we have now gone back to the study database to extract information about neonates and do some analysis.

To answer the question, neonates were neither excluded from the study nor were they grouped separately. 

Our observation about the neonates after been grouped has now been mentioned in detail in the result section in the text (line 239-242)

‘The additional age stratum that was not presented in figure and in table was neonatal age 0-4-week that account for 141/1427. Of these, there were 41 deaths of which ABM, SNBM and CSM account for 7/41, 4/41 and 30/41 respectfully’.

Line 120 – the non-bacterial definition – there is more to the diagnosis of “aseptic meningitis” than just culture and test negative – yet you have not defined lymphocytes vs neutrophils, other CSF parameters – I think it is okay to keep it simple – but then you should define as culture/pathogen positive vs culture/pathogen negative – as that is the only difference. I worry that some of your so-called aseptic are still bacterial meningitis, but were not cultured or detected with the methods you used (PCR may have detected pathogens). Also some of the clinically suspected could be “aseptic meningitis”? 

Your categories are really 

1. Culture positive/pathogen identified (irrespective of CSF findings?) 

2. Culture negative/pathogen not identified but abnormal CSF 

3. Clinically suspected with normal CSF 

The reviewer is correct and following the comments of reviewer 1 and 3 in addition to those of this reviewer the text has been modified to reflect this point.

Methods: how was serotyping and serogrouping performed? Please include in methods, with references if explained in more details elsewhere 

This has now been done and the following sentence included in the methods section of the text (line 169-176) 

‘serotyping for pneumococcal serotypes with a latex agglutination assay using factor and group-specific antisera (Statens Serum Institut, Copenhangen, Denmark). Neisseria meningitidis isolates were tested for serogrouping by latex agglutination using Ramel (Thermo Fisher Scientific, Waltman, MA, USA) test kits. H. influenzae type b and other encapsulated H. influenzae other than type b was by slide latex agglutination using polyvalent and monovalent Haemophilus influenzae Difco serotyping antisera to Hi of types a, b, c, d, e and f (Beckton Dickinson, Difco, Belgium)’.

Table 1 – for clarity see if you really need ABM, NBM and CSM abbreviations – as there may be enough space to write out? 

This has been done and the full wording included in all the tables including Table1.

Please consider defining these differently – so immediately useful to clinicians 

I wonder if Table 2a would be better as a figure, with CI?? 

We have considered this option but believe that data are clearer when presented in a table rather than in a figure. – The number involved is small for such analysis

Table 2b – this would also be useful by year, so each of the types of meningitis by age by year

Also, we have considered this option but prefer to keep it simple and clearer in this format – Again, the number involved is small for such analysis.

Table 3 – did you define “CSF” as CSF only or CSF and blood, so “blood” is only from blood – it looks like it as seem mutually exclusive, but we know you had cases with both CSF and blood – please define and clarify; 

We have grouped clinical samples in the study into three, namely; CSF only (only CSF sample was obtained from the patient), Blood only (only blood sample was obtained from the patient) and CSF and Blood (both CSF and blood were obtained from the patient). 

This was made clearer in the Figure 1 and now added in the method section in the text (line 150-152) 

‘Blood only or CSF only, or both blood and CSF, were collected from patient for conventional microbiological investigations.’

how does GNR (coliform; what does GNR stand for) differ from E.coli and Klebs – they are all coliforms, please correct and then any group of pathogens, use a footnote to list what that group contained. Your results are important and unique – please document the details. 

Thanks for this observation. We have now clarified this as suggested in the Table 4 and its footnote (line 334-338) as well as in Table 6 (line 447-459).

What is non-Hib – as in fact everything not Hib is non-Hib?? Or do you mean Haemophilus influenzae (non-typeable, or types a, c, d, e, and f?) 

Also, this is a helpful observation. 

‘Non-type b H. influenzae were H. influenzae isolates which did not agglutinate when tested with type b capsular antiserum. 

We have now made the changes in the Table 4 and Table 6 as well as throughout the text to 

‘non-type b H. influenzae’. Meaning H. influenzae other than type b (n=6; 5/6 of these were H. influenzae type a (line 331-332). 

Discussion: 10 years is a long time, were there other improvements in S/E conditions, general nutrition/health in the community? – as these are ecological data – can you talk about other admissions? Other diseases that may have not been affected by the vaccine? 

It is true that improvements in socio-economic conditions have occurred during the period of the study but data have been presented previously [Mackenzie et al. 2017 – Lancet Infect Dis 2017; 17: 965–73 http://dx.doi.org/10.1016/] that were from the same study area, the incidence of non-pneumococcal invasive infections did not decrease in line with the former

Please also compare to more of the African and international literature – where useful. It is not enough to say no other data from The Gambia or West Africa. 

This observation is well appreciated, we have searched for more literatures within Africa as suggested. Although this request would require a formal systematic review which is beyond the scope of this study. Nonetheless, the findings are similar to other studies conducted at about the same time in West Africa (Sidikou F 2019 and Darboe S 2019) now added in the discussion section in the text (line 555-556). 

‘This observation is consistent with other studies from sub-Saharan Africa associating meningitis with dry weather(references 16,27,33)’ 

One paragraph on improving diagnostics and the additional gain from PCR should be discussed in more detail – as your study does not show the added benefit (see line 323), or only quite indirectly really – so this needs to be discussed better. Consider below reference: 

Thanks a lot for taken note of this it is very useful. 

We have now complied to your suggestion by adding more information in the discussion section in the text (line 615-622) 

‘Access to appropriate diagnostics and effort in reducing diagnostic gaps is a significant global challenge. A recent study has described diagnosis as the biggest gap in the cascade of care caused by many factors[38]. The report corroborates findings from our study, underscoring the urgent need for newer molecular diagnostics[40]–[43] that better identify and distinguish viral, bacterial and fungal pathogens, to guide earlier and more appropriate clinical management of children with meningitis in sub-Saharan Africa. The time is now to employ molecular PCR approaches as routine investigation to detect pathogens from invasive and non-invasive clinical samples from ill children from sub-Saharan Africa.’

Lancet 2021; 398: 1997–2050 Published Online October 6, 2021 https://doi.org/10.1016/ S0140-6736(21)00673-5. The Lancet Commission on diagnostics: transforming access to diagnostics

Thanks a lot for calling my attention to the above publication. It is very helpful.

Line 326: antibiotics are not “emerging”, or do you mean as new antibiotics are introduced? 

Thanks. We have made the correction in the discussion section of the text (line 623-624)

‘emergence of antibiotic resistant-pathogens needs to be sustained to determine future trends in antibiotic resistance.’

Last paragraph – what do the authors think would be important interventions to decrease meningitis further. 

This would again require a separate paper but it is clear that increased access to diagnosis including rapid diagnostic tests at the district hospital level would help to prevent cases of meningitis being missed and increasing availability of PCR in more specialist hospitals would help to improve both overall clinical care and contribute to an improved understanding of changing aetiology and the epidemiology of ABM in a particular area. This has now been added to the discussion section of the text (line 633-652). 

‘We recommend increased access to rapid diagnostic and PCR tests across the hospital system to help prevent cases of meningitis being missed, improve both overall clinical care and contribute to an improved understanding of the changing aetiology and epidemiology of ABM in a particular area. Interventions to decrease meningitis further should aim to; introduce affordable pentavalent meningococcal conjugate vaccine against N. meningitidis serogroups ACWYX, and introduce higher valency pneumococcal conjugate vaccines such as PCV15 or PCV20 or PCV24 to provide wider protection against pneumococcal vaccine serotypes. Continuing surveillance will be required for monitoring trends in the evolution of serogroups and serotypes following introduction of new vaccines.’

.

Throughout use case-fatality ratio, and not interspersed with “rate” 7. 

Again thanks. We have made the correction to case fatality ratio throughout the text.

6. PLOS authors have the option to publish the peer review history of their article (what does this mean?). If published, this will include your full peer review and any attached files.

Do you want your identity to be public for this peer review? For information about this choice, including consent withdrawal, please see our Privacy Policy.

Reviewer #1: No

Reviewer #2: No

Reviewer #3: No

 Reviewer #4: No

---

## [Decision Letter · Decision Letter 1]

26 Jul 2022

Childhood meningitis in rural Gambia: 10 years of population-based surveillance

PONE-D-22-05903R1

Dear Dr. Ikumapayi,

We’re pleased to inform you that your manuscript has been judged scientifically suitable for publication and will be formally accepted for publication once it meets all outstanding technical requirements.

Kind regards,

Joël Mossong, PhD

Academic Editor

PLOS ONE

Additional Editor Comments (optional):

Reviewers' comments:

Reviewer's Responses to Questions

**Comments to the Author**

1. If the authors have adequately addressed your comments raised in a previous round of review and you feel that this manuscript is now acceptable for publication, you may indicate that here to bypass the “Comments to the Author” section, enter your conflict of interest statement in the “Confidential to Editor” section, and submit your "Accept" recommendation.

Reviewer #2: All comments have been addressed

Reviewer #3: All comments have been addressed

2. Is the manuscript technically sound, and do the data support the conclusions?

Reviewer #2: Yes

Reviewer #3: Yes

3. Has the statistical analysis been performed appropriately and rigorously? 

Reviewer #2: Yes

Reviewer #3: Yes

4. Have the authors made all data underlying the findings in their manuscript fully available?

Reviewer #2: Yes

Reviewer #3: Yes

5. Is the manuscript presented in an intelligible fashion and written in standard English?

Reviewer #2: Yes

Reviewer #3: Yes

6. Review Comments to the Author

Reviewer #2: Authors made all revisions and the recent version of the manuscript is suitable for publication, if other Reviewer's and Editorial Board also agree

Reviewer #3: The authors have made good approach at answering my main queries.

7. PLOS authors have the option to publish the peer review history of their article (what does this mean?). If published, this will include your full peer review and any attached files.

Reviewer #2: No

Reviewer #3: No

---

## [Editor Report · Acceptance letter]

2 Aug 2022

PONE-D-22-05903R1 

Childhood meningitis in rural Gambia: 10 years of population-based surveillance 

Dear Dr. Ikumapayi:

I'm pleased to inform you that your manuscript has been deemed suitable for publication in PLOS ONE. Congratulations! Your manuscript is now with our production department. 

Kind regards, 

on behalf of

Dr. Joël Mossong 

Academic Editor

PLOS ONE